# *Citri grandis* Exocarpium Extract Alleviates Atherosclerosis in ApoE^−/−^ Mice by Modulating the Expression of TGF-β1, PI3K, AKT1, PPAR-γ, LXR-α, and ABCA1

**DOI:** 10.3390/foods14244267

**Published:** 2025-12-11

**Authors:** Jing Xu, Wen-Zhao Wen, Jun-Hui Zhao, Jun-Rong Guo, Zhuo-Ya Zhang, Ping Xiong

**Affiliations:** Pharmaceutical Engineering Specialty, South China Agricultural University, Guangzhou 510640, China; 17898488256@163.com (J.X.); 20223138216@stu.scau.edu.cn (W.-Z.W.); 18122454201@163.com (J.-H.Z.); 13556181173@163.com (J.-R.G.); zzy20233185085@stu.scau.edu.cn (Z.-Y.Z.)

**Keywords:** *Citri grandis* exocarpium, atherosclerosis, lipid metabolism regulation, reverse cholesterol transport

## Abstract

*Citri grandis* exocarpium (*Citri grandis*) has been consumed by human beings for fifteen hundred years. It is commonly consumed as a health drink and dietary supplement in China. However, its nutritional and healthcare functions are still not fully understood. **Objective:** Our previous study found that oral administration of *Citri grandis* extract can significantly decrease the blood lipid levels of hyperlipidemic mice fed a high-fat diet. The aim of this study was to confirm the preventative effects of *Citri grandis* extract against atherosclerosis. **Methods:** Atherosclerotic lesion models were induced in HUVECs and apoE^−/−^ C57BL/6J mice. ApoE^−/−^ mice fed a high-fat diet were orally administered *Citri grandis* extract (0.4, 0.8, and 1.6 g/kg/d BW) and Simvastatin (1 mg/kg/d BW) on the first day of model establishment. After a 16-week treatment, serum samples and aorta and liver tissues were collected. Observation of pathological changes in aortic and liver tissues was performed using a light microscope with oil red O, H&E, Masson’s trichrome staining, and TEM. Biochemical detection was employed to determine the serum levels of TC, TG, LDL-C, and HDL-C as well as the activities of AST and ALT. In addition, expression studies of TGF-β, PI3K, AKT1, PPAR-γ, LXR-α, and ABCA1 were performed via qPCR and Western blot analysis. **Results:** Compared with cholesterol-induced HUVECs, *Citri grandis* extract significantly enhanced cell viability, attenuated the morphological changes in HUVECs, and reduced LDH release. Furthermore, after treatment with *Citri grandis* extract, the levels of TC, TG, and LDL-C significantly decreased in the atherosclerosis model apoE^−/−^ mice after 16 weeks, and aortic plaque, lipid deposition, and endothelial injury were obviously ameliorated. The mRNA and protein expression of TGF-β, PPAR-γ, LXR-α, and ABCA1 in aortic and liver of atherosclerosis apoE^−/−^ mice were upregulated (*p* < 0.05, *p* < 0.01), while those of PI3K and Akt1 were suppressed (*p* < 0.05, *p* < 0.01). **Conclusions:** *Citri grandis* extract can significantly decrease the high circulating lipid levels and the liver lipid deposition of high-fat-diet-fed apoE^−/−^ mice and reduce aorta lipid accumulation and atherosclerotic plaques by regulating the expression of TGF-β1, PI3K, AKT1, PPAR-γ, LXR-α, and ABCA1. *Citri grandis* extract can be used as a healthcare dietary supplement for the prevention of abnormal lipid metabolism and atherosclerosis.

## 1. Introduction

*Citri grandis* exocarpium (*C. grandis*), belonging to the *Citrus* genus in the family *Rutaceae*, is the nearly mature or immature dried exocarp of this species. *C. grandis* obtained official approval for use as a homologous plant in medicine and food in August 2024 in China. *C. grandis* has alternatively been called *Citrus grandis* “Tomentosa,” *Citrus grandis* (L.) Osbeck, *Citrus maxima* “Tomentosa,” etc., and as it originates from Huazhou, China, it is also called Hua Ju-Hong.

*C. grandis* has been consumed by humans for fifteen hundred years, and its earliest cultivation can be traced back to the “Liang” dynasty of China. The earliest record of *C. grandis* was found in “Gaozhou fuzhi” in the WanLi period of the Ming dynasty, and it has also been recorded in subsequent works of the Chinese materia medica, such as “Bencao Gangmu Shiyi,” “Bencao yuanshi,” “Bencao congxin,” “Bencao Fengyuan,” and so on, up to the current version of the Chinese Pharmacopeia [1]. *C. grandis* is considered an important drug with antitussive and expectorant effects [2,3].

The literature on *C*. *grandis* has mainly focused on cultivation techniques, species identification, origin investigation [4], its antitussive and expectorant activities [2], and its chemical composition [5], with flavonoids [6,7], volatile oil [8,9], polysaccharide [10], coumarin [11,12], inositol [13,14], and particularly naringin, naringenin, and rhoifolin being the main active ingredients [1,15]. However, there is still much more to learn about its nutritional and healthcare functions.

Our previous study confirmed that *C. grandis* extract (CGE) could significantly decrease the blood serum levels of cholesterol and triglycerides and alleviate hepatic steatosis and liver lesions [16]. Recent studies have revealed that lipid metabolism dysfunction is a vital risk factor related to the formation and development of atherosclerosis (AS) [17,18]. Consequently, we speculated that CGE has certain effects pertaining to the prevention of AS and the improvement of blood vessel endothelium dysfunction.

AS is a slowly progressive inflammatory disease and the underlying cause of myocardial ischemic attack and stroke [19]. Many studies indicate that macrophages play a critical pathogenic role in the formation of atherosclerotic lesions [20,21]. Recent studies have strongly suggested that the immune inflammatory response involving macrophages influences the formation and development of AS, and the PI3K/Akt1 pathway can participate in the regulation of the AS process via the influence of macrophage polarization, lipid metabolism, and autophagy [19,22]. As is well known, TGF-β can induce multiple types of cells, including endothelial and epithelial cells. Together with the PI3K/Akt1 signaling pathway, it modulates these cells’ proliferation, apoptosis, and migration. It is possible that systemic administration of PI3K/Akt1 inhibitors may be associated with increased apoptosis in macrophages and, during the early stages of AS, may suppress lesion growth [23]; therefore, they are a potential target for the prevention of AS. In addition, although there are multiple mechanisms involved in the efflux of cellular cholesterol, recent studies have indicated that the ABCA1 mediates this process and is involved in the control of the HDL and apoAl-mediated cholesterol efflux from macrophages [24]. PPAR γ was recently shown to induce the expression of the cholesterol transporter, ABCA1, in macrophages through a transcriptional cascade mediated by the nuclear receptor, Liver X receptor α (LXR-α) [25]. Taken together, PPAR-γ exerts anti-atherogenic effects by facilitating the removal of cholesterol from macrophages via cholesterol transporter proteins such as ABCA1.

In this study, we investigated the effect of CGE on some proteins, including ABCA1, which mediates cholesterol efflux, and TGF-β, which is associated with the development of atherosclerotic lesions. We hope to provide more evidence that CGE is a healthy food supplement suitable for use in the prevention of hyperlipidemia and AS.

## 2. Materials and Methods

### 2.1. Plant Material and Chemical Reagents

CGE was provided by Xiangxue Pharmaceutical Co., Ltd. (Guangzhou, China). PIPA lysis buffer, phospholipase inhibitors, protease inhibitor, 5× protein loading buffer, sodium dodecyl sulfate (SDS), glycine, PMSF, Tris base, Hifair^®^ II 1st Strand cDNA Synthesis Kit, LDH detection kit, fetal bovine serum (FBS), Cell/Tissue Total RNA Kit, DL2000 DNA Marker, protein marker, ECL luminescence reagent, Goldview nucleic acid gel staining (10,000×), DMEM medium with high glucose, qPCR SYBR Green Master MixPCR, and the Hifair^®^ II 1st Strand cDNA Synthesis Kit were purchased from Yeasen Biotechnology Inc. (Shanghai, China). LDL-C, HDL-C, TC, TG, GOT, and GPT detection kits and BCA protein assay kit were purchased from Nanjing Jiancheng Bioengineering Institute (Nanjing, China). Monoclonal antibodies to GAPDH (rabbit) were obtained from AB Clonal (Wuhan, China). Polyclonal antibodies against TGF-β1 (rabbit) and goat anti-rabbit secondary antibodies were obtained from UpingBio Technology Co., Ltd. (Hangzhou, China). Polyclonal antibodies against PI3K p85 (rabbit) were obtained from Affinity Biosciences Co., Ltd. (Liyang, Jiangsu, China). Polyclonal antibodies for AKT1 (rabbit), PPAR-γ (rabbit), LXR-α (rabbit), and ABCA1 (rabbit) were obtained from Proteintech (Wuhan, China). A primary antibody dilution buffer was purchased from Yeasen Biotechnology Inc. (Shanghai, China). A 0.45 um PVDF membrane was purchased from Millipore (Billerica, MA, USA). DMSO (Shanghai Macklin Biochemical Technology Co., Ltd., Shanghai, China), Trypan Blue Stain (SIGMA-ALDRICH, Trading Co., Ltd., Shanghai, China), 0.25% Trypsin-EDTA, penicillin–streptomycin antibiotics (Gibco, Grand Island, NY, USA), MTT, PBS buffer dry powder (Wuhan Boster Biological Technology Co., Ltd. Wuhan, China), 30% Acr-Bis, 10% APS, TEMED, and 1 × TBST buffer were purchased from Solarbio (Beijing, China). The 50×TAE, cholesterol (Yong Jin Biotech Co., Ltd, Guangzhou, China), oil red O dye (Sangon Biotech Inc., Beijing, China), Naringin (No. 110722-201815), and rhoifolin (No. 111919-201804) were from the National Institutes for Drug and Food Control (Beijing, China). Meranzin hydrate (No. Y29J9H64677), and isomerazin (No. W10S9Z69771) were purchased from Shanghai yuanye BioTechnology Co., Ltd. (Shanghai, China). With the exception of methanol, acetonitrile (HPLC grade, Merck Chemical Co., Ltd., Shanghai, China), and glacial acetic acid (HPLC grade, Aladdin Reagent Co., Ltd. Shanghai, China), all reagents were of analytical pure grade.

### 2.2. Cell Lines and Animals

The HUVEC strain was donated by Professor Yang Chen from Guangzhou University of Traditional Chinese Medicine, China. A total of 60 8-week-old apoE^−/−^ male C57BL/6J mice with 18 ± 2 g body weight were provided by JunKe Biological Co., Ltd. (Nanjing, China), with the license number SCXK (Su) 2020-0009. Another 12 8-week-old male C57BL/6J mice with 18 ± 2 g body weight were purchased from JunKe Biological Co., Ltd. (Nanjing, China) (SCXK (Su) 2020-0009). All mice were kept in stainless steel cages and in consistent environmental conditions (temperature: 23 ± 1 °C; light/dark cycle: 12/12 h; relative humidity: 40–70%). Mice were fed with complete formula feed and were provided with water ad libitum. All animal experiments were approved by the Animal Ethics Committee of South China Agricultural University (Approval No. 2024G039). All procedures followed the Guide for Care and Use of Laboratory Animals published by NIH, Bethesda, MD, USA [26].

### 2.3. Preparation of CGE

CGE was prepared according to the optimal extraction conditions described previously [17]. In brief, *C. grandis* was extracted with filtered water through reflux extraction (1:30, 70 °C) three times, each time for 1 h. All extracted liquid was merged, filtered, and then concentrated under reduced pressure at 60 °C with a rotary evaporator (Buchi, Frauenfeld, Switzerland). Finally, the concentrated liquid was dried with a spray dryer (Attainpak Equipment, Shanghai, China) to obtain a dry powder. A total of 1 g of CGE is equivalent to 3.42 g of *C. grandis*.

### 2.4. HPLC Fingerprint Procedure

An HPLC instrument from the Waters alliance e2695 HPLC system (Waters Corporation, Milford, MA, USA), equipped with a low-pressure quaternary pump, an online vacuum degas system, a thermostatic auto-sampler, a DAD, and an Empower 3 with Waters Chemstation software (Empower 2), was used to obtain the HPLC chromatographic fingerprints. Chromatographic separations of CGE were performed on a C_18_ analytical column (Waters Symmetry, 4.6 mm × 250 mm inner diameter, 5 μm particle size) supplied by Waters Corporation (Milford, MA, USA).

The mobile phase was composed of a mixture of A (0.1% acetonitrile acetate)-B (0.1% acetic acid solution). Gradient elution with the ratio of A:B, varied as below: 0–20 min, 5–22% A; 20–50 min, 22–42% A; 50–70 min, 42–100% A; 70–75 min, 100% A; 75–78 min, 100–5% A. The input sample volume was 10 μL, the column temperature was maintained at 30 °C, and the mobile phase flow rate was 0.6 mL·min^−1^. The sample was filtered through a 0.22 μm syringe filter (PTFE). All solvents were filtered through a 0.45 μm Millipore filter before use. Chromatographic fingerprints were recorded at a detection wavelength of 320 nm.

### 2.5. Standard Solution and Sample Solution Preparation

We accurately weighed an appropriate amount of naringin and rhoifolin with MS204S electronic balance (Mettler Toledo, Zurich, Switzerland) and then added methanol to prepare a mixed standard solution with 1600 μg/mL of naringin and 50 μg/mL of rhoifolin. After that, 1 mL of mixed reference solution was precisely taken and then diluted 2-, 5-, 10-, 25-, and 50-fold with methanol, respectively, to prepare different concentration standard solutions. In addition, we accurately weighed 0.50 g of CGE powder with an MX5 Mettler Toledo electronic balance (Zurich, Switzerland) and put it in a conical flask with stopper, after adding 25 mL of ultrapure water, which was weighed and shaken well. Finally, 2.00 g of the diluted solution was weighed precisely, transferred to a 10 mL volumetric flask, and then diluted with methanol to volume. The sample solution was obtained after being shaken well and filtered.

### 2.6. Cell Viability Assay

The viability of HUVECs was assessed by MTT assay. In brief, the HUVECs (1 × 10^4^) were inoculated in a 96-well cell culture plate with DMEM (10% FBS) in a final volume of 0.2 mL. We allowed cells to attach at 37 °C. When the cell density reached 80% confluence, they, respectively, were treated with CGE at two-fold serial concentration and incubated at 37 °C in a 5% CO_2_ for 24 h. Cell blank and control groups were set up. Each group has at least six parallel wells. After the indicated treatments, HUVECs were washed with PBS, and 22 μL of 5 mg/mL MTT was added to each well. After the incubation for 4 h, the formazan of each well was completely dissolved in 150 µL of 100% DMSO solution at room temperature. The OD values of all groups at the 570 nm wavelength were determined with a Cytation^TM^ 5 microplate reader (Biotek, Instruments, Inc., Winooski, VT, USA), and HUVECs’ viability was calculated according to the formula. The experiment was repeated three times.

### 2.7. LDH Release Assay

The protection of CGE to HUVECs from injury was investigated through the release level of cytosolic LDH. In brief, the treatment method was the same as previously described. HUVECs were divided into six groups, with six parallel wells for each group, which included a control group, a blank group, a model group, and high-, medium-, and low-dose CGE groups. In this test, HUVEC injury was induced by 10 μmol/L of cholesterol, and 5.00, 2.50, and 1.25 mg/mL concentrations were used as the treatment dose of CGE. The viability of HUVECs was measured as previously described.

In addition, after the cholesterol-induced HUVECs were treated with 1.25, 2.50, and 5.00 mg/mL of CGE for 24 h, they were incubated for 1 h with 150 μL of cell lysis buffer in each well. The supernatants were collected via centrifugation at 400× *g* at 4 °C for 5 min and transferred to another new 96-well plate with 120 μL in each well. In the end, LDH release was determined with an LDH assay kit according to the manufacturer’s instructions. A total of 60 μL of LDH working solution was added into each well; then, the 96-well plate was incubated in the dark for 30 min. LDH activity was determined via a colorimetric assay using a wavelength of 490 nm in a spectrophotometer (Bio-Rad Laboratories). We can use any one of the wavelengths at 600 nm or above as a reference. The experiment was performed independently three times. The rate of cell damage was calculated according to the OD values.$$Cell\;damage\;rate\;(\%)=\frac{A_{1}-A_{0}}{A_{2}-A_{0}}\times 100\%$$
where *A*_0_ is the absorbance of control (distilled water instead of the sample solutions); *A*_1_ is the absorbance of the tested samples; and *A*_2_ is the absorbance of the maximum enzyme activity in the cells.

### 2.8. Oil Red O Staining for Cholesterol-Induced HUVECs

Cell concentration was adjusted to 2.5 × 10^5^ cells/mL. A total of 2 mL of cell suspension was inoculated in each well of a 6-well plate and cultured for 24 h to attach. Intracellular lipids were determined by the ORO-staining method with a slight modification [27,28]. The Lipid accumulation of HUVECs was assayed using an inverted microscope with Image J software (version: 1.54j, NIH, Bethesda, MD, USA).

### 2.9. In Vitro Cell Migration and Invasion Assay

The scratch assay was used to investigate cell migratory ability. HUVECs (5 × 10^5^/well) in 6-well plates were cultured in DMEM with 10% FBS for 24 h in a CO_2_ incubator at 37 °C. When the cells reached 90% confluence, they were cultured with complete medium containing 1–2% FBS in a final volume of 2 mL for 24 h. After that, scratch areas were created by p200 pipet tips, and cells were induced with 50 µg/mL ox-LDL and treated with 1.25, 2.50, and 5.00 mg/mL of CGE. The photoshops at 0 and 48 h after scratching were taken under an inverted microscope (ECLIPSE Nikon, Tokyo, Japan), and the scratch areas were measured using the Image J software (version: 1.54j, NIH, Bethesda, MD, USA).

In addition, the transwell assay was used to assess the impact of CGE on the invasiveness of HUVECs. In brief, the HUVECs (3 × 10^5^/well) were inoculated in a 6-well cell plate and treated with 50 µg/mL ox-LDL and 1.25, 2.50, and 5.00 mg/mL of CGE (final concentration in the well) for 24 h at 37 °C. In addition, the transwell was coated with 0.1% gelatin for 30 min at 37 °C, followed by washing with PBS three times. Then, cell suspensions were incubated in the upper chamber of a 24-well transwell chamber (JET BIOFIL Co., Ltd., Guangzhou, China) at a density of 5 × 10^4^/well with DMEM in a final volume of 0.1 mL per well, while the bottom chambers were filled with DMEM with 10% FBS. After the incubation for 24 h at 37 °C, cells on the upper side of the membrane were removed, and cells on bottom side were seen as the invasion cells. Then, the invasion cells were fixed with 4% PFA at RT for 30 min. The invasion cells were washed with PBS three times and then stained with 0.1% crystal violet solution for 15 min. Finally, random five fields of view per well were acquired under a microscope (ECLIPSE Nikon, Tokyo, Japan), and the number of positive cells in each field of view was counted using the Image J software (version: 1.54j, NIH, Bethesda, MD, USA). Each experiment was performed independently three times.

### 2.10. Modeling and Grouping

AS is known as a chronic inflammatory disease, which is initiated in most cases by hypercholesterolemia. The mouse has been recognized as the best model for experimental AS, and the apoE^−/−^ model has been found to develop extensive atherosclerotic lesions when fed a chow diet [29]. Briefly, a total of 60 18 ± 2 g weight and 8-week-old apoE^−/−^ male mice were divided into five groups randomly after being fed with a regular diet for 1 week. These groups included the model group (*n* = 12), the Simvastatin group (*n* = 12), and three CGE dose groups (*n* = 36). The high-fat diet had 21% fat and 0.15% cholesterol but was otherwise a normal diet. The C57BL/6 male mice were fed with a normal diet as normal control group (NC; *n* = 12). The Simvastatin group and the three CGE dose groups were administered via oral gavage with Simvastatin solution (1 mg/kg) and CGE solution (1.6, 0.8, and 0.4 g/kg · BW), respectively, once per day for 16 weeks. The administration doses were calculated based on 60 kg adults using a human equivalent dose formula. The NC group and MOD group were administered via oral gavage with equal volumes of filtered water. All procedures followed the Guide, USA [26].

### 2.11. Sample Collection and Preparation

At the end of the experiment, 1.2–1.5 mL of blood was collected from the inner canthal orbital vein of the mouse and was then centrifuged at 12,000 rpm for 15 min to separate the serum. They were stored in a −80 °C refrigerator for the biochemical assay. After that, mice were euthanized by i.p. injection of 0.35% pentobarbital sodium solution (0.1 mL·10^−1^ g BW), and their aortic blood vessels and livers were taken out and rapidly rinsed with cold PBS and lightly patted dry with filter paper. Finally, the aortic and liver tissues of each group were divided into small pieces on the ice platform. Each piece was sub-packaged, weighted, marked, and stored at −80 °C.

### 2.12. Biochemical Detection

Four serum lipids, AST, and ALT were detected by assay kits (Nanjing Jiancheng Bioengineering Institute, Nanjing, China) following the manufacturer’s instructions. The standard curve of ALT and AST activity was established by a microplate reader (Biotek, Winooski, VT, USA), with optical density values at a 510 nm wavelength. The blood lipid levels and active AST and ALT units for the different groups were determined according to the OD values and the relevant calculation formulae.

### 2.13. Oil Red O Lipid-Staining Assay

The quantification of the ORO-stained atherosclerotic lesions was performed using basic morphometric methods to assess the burden [30]. After isolation of the whole aorta, the accumulation of lipids on the aortic vessel wall was observed by ORO staining. Briefly, the fresh aortic vessels were cut open along the long axis after removing external fat around aorta under an MZ101 stereomicroscope (Mingmei Photoelectric Technology Co., Ltd., Guangzhou, China); then, aortic tissues were fixed in 4% PFA solution for 2 h at RT. Before staining, the aortas were washed with filtered water three times for 5 min each time. After the incubation of 60% ethanol for 20–30 s, the samples were further incubated with oil red O (Sangon Bio-tech Co., Ltd., Shanghai, China) solution at 60 °C in the dark for 10–15 min. A total of 85% propylene glycol removed the excess ORO solution, and finally, samples were rinsed with distilled water three times. The stained aortas were spread out on a black background and photographed under a stereomicroscope. Semi-quantification was carried out using the following formula:$$The\;lipid\;content=\frac{ORO\;stained\;area}{Lumen\;area}$$

### 2.14. Histopathology Examination and Collagen Deposition Evaluation

Histopathological examination of mouse liver and aorta tissues was performed via an optical microscope and TEM. The tissues were fixed in 4% neutral-buffered PFA solution for 48 h; then, tissue sections were embedded with paraffin. After that, the tissue slices were subjected to ORO, Masson’s trichrome, and H&E staining at room temperature. Finally, the stained slices were observed through a microscope (Olympus, Tokyo, Japan) and photographed using a Nikon 4500 digital camera (Nikon, Tokyo, Japan). In addition, the aortas underwent TEM observation. Briefly, mouse aortas were fixed with 2.5% glutaraldehyde and post-fixed with 3% osmium tetroxide (OsO_4_) for 2 h. The specimen was dehydrated in series of graded ethanol, embedded in epoxy resin, and then sliced into 100 nm pieces, which were stained with double lead, and finally imaged with a TEM at 80 kV (HT7800/HT7700 Hitachi, Tokyo, Japan). The semi-quantification was carried out using the following formula: $$The\;collagen\;fiber\;content=\frac{Area\;of\;Masson’s\;trichrome-stained\;collagen}{Lumen\;area}$$

### 2.15. Gene Expression Analysis via RT-qPCR

The fresh aorta and liver tissues were lysed with tissue lysis buffer (LB) for 3 min in an ice bath; then, the lysis solution was homogenized with an ultrasonic device (100 W for 20 s × 4 times). The lysis solution was centrifuged at 14,000 rpm for 15 min at 4 °C to obtain the supernatant. Then, the total RNA was extracted from the supernatants by a cell/tissue total RNA kit following the instructions. Afterwards, the RNA was dissolved in 20 μL RNase-free ddH_2_O (Qiagen, Düsseldorf, Germany). The concentration of mRNA was quantified with a NanoDrop^TM^ND-1000 device (Thermo Fisher Scientific, Waltham, MA, USA). Total RNA was reverse-transcribed into cDNA using the reverse-transcriptase kit (Yeasen Biotechnology, Shanghai, China). The cDNA synthesis was performed following the instructions by mixing 5 μg RNA, 1 μL g DNA digester, and 2 μL 5× g DNA digester buffer and finally adding RNase-free ddH_2_O to a 10 μL final volume. PCR reaction conditions were 5 min at 25 °C, 30 min at 45 °C, 5 min at 85 °C, and 4 °C indefinitely. In the end, the quantitative PCR was performed using a Real-Time PCR System (CFX96TM C1000, Bio-Rad, Hercules, CA, USA). The primers were designed using Primer Premier 5 and NCBI Primer-BLAST software system (San Diego, CA, USA). The specific primers are shown in Table 1. The q-PCR reaction was carried out in 20 μL volume using an optical 96-well tray according to the instructions of qPCR SYBR Green Master Mix kit. The reaction system consisted of 10 μL of MasterMix (Yeasen Biotechnology, Shanghai, China), 0.4 µL of each primer (10 µM), 2.0 µL of cDNA template, and finally, RNase-free H_2_O was added up to 20 µL in the PCR tubes. Each sample was analyzed in quadruplicate. The mRNA expression levels were quantified by measuring the threshold cycle (CT) values of the genes.

### 2.16. Western Blot Analysis

The protein expression levels of PI3K, TGF-β, AKT1, PPAR-γ, ABCA1, and LXR-α in mouse aorta and liver tissues were assayed. The tissues were lysed using RIPA lysis buffer (Yeasen Biotechnology Co., Ltd. Shanghai, China) for 15 min in an ice bath, followed by centrifugation at 12,000 rpm at 4 °C for 30 min. The protein was quantified using the BCA method, then 5× loading buffer was added at a ratio of 4:1. Afterward, the mixture was denatured with boiling water for 5 min. The proteins were separated by 12% SDS-PAGE gel at 70 V for 30 min and then 120 V for 90 min and were then transferred to 0.45 μm PVDF membranes (Merck Millipore, Billerica, MA, USA) via electro-blotting according to a wet transfer protocol (270 mV, 90 min). Blots were blocked with 20 mL of 1% BSA in Tris-buffered saline with 0.1% Tween-20 at RT on a 60 rpm shaker for 2 h and then rinsed six times with 1×TBST buffer on the shaker at 90 rpm, 5 min each time. Afterwards, the membranes were incubated with primary antibodies against PI3K P85 (rabbit, 1:500), TGF-β (rabbit, 1:500), AKT1 (rabbit, 1:500), PPAR γ (rabbit, 1:500), LXR-α (rabbit, 1:500), ABCA1 (rabbit, 1:500), and GAPDH (rabbit, 1:50,000) (Abcam, Cambridge, UK) at 4 °C overnight, followed by incubation with HRP-labeled goat anti-rabbit secondary antibodies (1:15,000) for 1 h at RT after repeating the previous wash three times. Finally, the immuno-reactive bands were documented using an enhanced chemiluminescent WB system (Thermo Fisher Scientific, Waltham, MA, USA) on a VersaDoc imaging system. Densitometric quantification of band intensity was performed with Image-Pro plus 6.0 software, and the integrated absorbance (IA) of the protein bands equals the mean gray value × area.$$The\;target\;protein\;expression\;level=\frac{IA\;of\;the\;target\;protein}{IA\;of\;the\;reference\;protein}$$

### 2.17. Statistical Analysis

The SPSS statistical Software package (SPSS Inc., San Diego, CA, USA) with Windows Version 20.0 was applied for statistical analyses of the experiment data. The statistical results are presented as $\bar{x}$ ± SE. The comparison of the measured variables between groups was performed using the MANOVA method combined with a Wilk’s Lambda, etc., if the variance is homogeneous. A Shapiro–Wilk test for multiple comparisons should follow this analysis; if not, we transform the data to a base 10 logarithm or rank. If the dependent variables are completely irrelevant to each other, One-Way ANOVA method was used, followed by appropriate post hoc tests. *p* < 0.05 indicated that the difference was statistically significant, and *p* < 0.01 indicated that the statistical difference was extremely significant. GraphPad Prism 5 software (GraphPad Software, Inc., San Diego, CA, USA) was used for plotting.

## 3. Results

### 3.1. Phytochemical Characterization via HPLC Analysis

The compositions of CGE were analyzed by HPLC chromatography. The conveient HPLC method established was validated for linearity, sensitivity, repeatability, and accuracy. The main characteristic chemical compositions for CGE were naringin, rhoifolin, meranzin hydrate, and isomerazin. The standard curve was drawn with the concentration of reference solution as the *x*-axis (X), and the peak area of the reference spetrum as the *y*-axis (Y). Through linear regression analysis, the linear regression equation for the content of rhoifolin is Y = 34256 X + 638.65 (γ = 0.9999), and that of naringin is Y = 5225.1 X + 16157 (γ = 0.9999). Based on the regression equation combined with the peak area of the chromatogram spectrum, the content of naringin and rhoifolin in this batch of *C. grandis* dried extract is 96.34 mg/g, and 8.86 mg/g respectively. The relative retention times for naringin and rhoifolin were 31–32.5 min (Figure 1). 

### 3.2. CGE Prevents Cholesterol-Induced Endothelial Injury

Initially, the possible cytotoxic effects of CGE on HUVECs were detected. The results found no significant influence on cell viability after 24 h of incubation with 1.25, 2.50, and 5.00 mg/mL of CGE, while 20 mg/mL of CGE caused an increase in the cell cytostatic rate from 0% to 92.44 ± 0.67% (Figure 2A). In addition, 2.50, 5.00, 10, and 20 μM cholesterol incubation for 24 h resulted in an obvious dose-dependent decrease in cell viability (Figure 2B). Moreover, there was only 47.82 ± 1.81% cell viability after incubation with 20 μM cholesterol. After pretreatment with 5.00, 2.50, and 1.25 mg/mL of CGE for 24 h and incubation with 10 μmol/L cholesterol, it was found that 1.25, 2.50, and 5.00 mg/mL of CGE obviously enhanced the viability of HUVECs treated with cholesterol to 74.47 ± 5.38%, 78.98 ± 5.28%, and 79.35 ± 3.83%, respectively, which represented a significant deference compared with the 59.60 ± 1.44% of the model group induced with cholesterol (*p* < 0.05, *p* < 0.01) (Figure 2C). We assayed the effect of CGE on LDH release. Compared with the model group, CGE pretreatment with 1.25, 2.50, and 5.00 mg/mL significantly attenuated LDH release in HUVECs (Figure 2D), and the LDH release rate of HUVECs decreased to 35%, 30%, and 20%, respectively. As a result, CGE helped protect HUVECs against cholesterol-induced injury.

Figure 2E shows different degrees of red lipid droplets in the cholesterol-induced HUVECs after treatment. The red lipid droplets of NC group were almost invisible. The lipid accumulation of MOD group was severe, and large red lipid droplets were noted. Compared with the MOD group, CGE significantly reduced the lipid droplet accumulation in HUVECs and in a dose-dependent way (*p* < 0.01, *p* < 0.001), suggesting that CGE prevented HUVECs from lipid deposition formation.

Based on the cell viability results, the concentrations of CGE at 1.25, 2.50, and 5.00 mg/mL were selected for the anti-proliferative experiments. As shown in Figure 2F,H, for the HUVECs stimulated by ox-LDL, the prepared scratch by the pipette tip was almost fully filled with HUVECs. However, after treatment with CGE at 1.25, 2.50, and 5.00 mg/mL, the cell migration distances of the stimulated HUVECs were significantly suppressed (*p* < 0.01, *p* < 0.001, *p* < 0.001), in a dose-dependent relationship, compared to the control cells (HUVECs treated with ox-LDL). This result shows that CGE could inhibit cell migration in HUVECs.

In addition, the anti-invasive effect of CGE was determined. As shown in Figure 2G,I, the results indicate that the cell invasion capability of HUVECs was greatly enhanced (*p* < 0.001, vs. normal HUVECs control) after being stimulated by ox-LDL. However, CGE at 1.25, 2.50, and 5.00 mg/mL could obviously reduce the cell-invasive capability of the HUVECs stimulated by ox-LDL (*p* < 0.001).

### 3.3. Serum Lipid Profile and Transaminase Levels

There was no statistical difference in initial body weight among all groups, as is shown in Figure 3A, and the body weight of the HFD-fed apoE^−/−^ mice groups increased faster than that of the NC group, indicating that feeding a high-fat diet can increase body fat, particularly during the period of the 30th to the 100th day—in fact, the body weight of all the groups continuously increased throughout the whole experimental period; however, the body weight of the MOD group had no obvious difference from that of the other groups (*p* > 0.05).

In addition, the serum levels of LDL-C, TG, and TC in the MOD group significantly increased (*p* < 0.01), while the serum level of HDL-C showed no statistical difference (*p* > 0.05) when compared with the NC group. The apoE^−/−^ male mice in the CGE and Simvastatin groups exhibited obviously lower serum levels of TG, LDL-C, and TC than the MOD group (*p* < 0.01, *p* < 0.05), indicating that CGE has a dual effect on TC and TG reduction. Moreover, the differences in the effect between the Simvastatin group and the CGE group were non-significant (*p* > 0.05). Detailed results are illustrated in Figure 3B.

In the model group, the serum levels of ALT and AST increased significantly when compared with the NC group (*p* < 0.01). The apoE^−/−^ mice in the CGE group and the Simvastatin group exhibited obviously lower serum levels of ALT and AST than the MOD group (*p* < 0.01). The detailed results were presented in Figure 3C.

### 3.4. Pathological Examination of Mouse Aorta

ORO staining was used to determine the anti-atherosclerosis effects of CGE on HFD-fed apoE^−/−^ mice. Atherosclerotic plaque formation was seen in the aortic root area. As shown in Figure 4A,B, there is no obvious lipid accumulation in the NC group, while the MOD group showed an increased amount of ORO-stained lipids with aorta intimal thickening (*p* < 0.001) in comparison with the NC group. The amount of ORO-stained lipids in the Simvastatin group and the CGE group decreased obviously in comparison to the MOD group (*p* < 0.05, *p* < 0.01).

Blue-stained collagen fibers commonly reflect the degree of tissue fibrosis. In the MOD group, the AS plaques had increased collagen fibers, as determined by Masson staining (*p* < 0.001), in comparison to the NC group. In the Simvastatin and CGE groups, the content of collagen fiber in the atherosclerotic plaques decreased homogenously in comparison with the MOD group (*p* < 0.001, *p* < 0.05, *p* < 0.01). Detailed data are presented in Figure 4C. As shown in Figure 4D, there are no obvious AS plaques in the NC group. In the MOD group, a great amount of AS plaque protruded from the aortic intima, the cavity became narrow, and the integrity of aortic intima was compromised. Compared with the MOD group, the areas of AS plaque in the Simvastatin group and the CGE group were obviously reduced. The vessel walls of the Simvastatin group became thick and had obvious lipid deposition in intima, with intact vessel endothelium and local roughness but no atherosclerotic necrosis. The aortic endothelium of the CGE-L group had partial deletion and less lipid accumulation, with a few foam cells under the intima, and without necrotic fibrous plaque. The CGE-M and CGE-H groups had intact vessel endothelium with intact fiber caps at the top of protuberant plaques (stable plaques). Compared with the CGE-L group and the MOD group, the lipid accumulation and foam cells under the intima were obviously reduced.

As shown in Figure 4E, TEM revealed that the cross-section of the apoE^−/−^ mice aorta in the MOD group had the typical characteristic of atherosclerotic plaques, the permeability of endothelial cell membrane had increased, and the bilayer structure of the nuclear membrane had disappeared. There was a cavitation of mitochondria within the endothelial cells and a vast amount of visible cavities or cholesterol calcium salt crystals between the endothelial cells and the internal elastic membranes, with a broken aortic endothelium. Compared with the MOD group, the NC group had no obvious atherosclerotic lesions and aortic calcification, with visible local dissolution of nuclear membrane in the endothelial cells. However, in comparison to the MOD group, the aortic damage in the Simvastatin group and the CGE group was apparently relieved. The lesions on the aortic endothelial cells in the Simvastatin group were reduced, with an intact aortic wall, though the fibrous tissue proliferation was visible in the sub-endothelial layer. CGE-M group had a few cholesterol crystals within the basement membrane of the endothelial cells, with a discontinuous endothelial cell layer. Some lipid particles of the endothelial cells were observed in the CGE-L group, with thickened membranes of the endothelial cells and visible calcium salt depositions in the intima lower layer. In the CGE-H group, aortic endothelial cell contraction appeared, with less deficiency in the cells’ basement membranes and a few depositions of cholesterol crystals in the lower layer of endothelium.

### 3.5. Pathological Observation of Mouse Livers

The lipogenesis rate and steatosis of mouse livers were investigated through visual observation. As shown in Figure 5A, the MOD group had liver lipid denaturalization and granular degeneration, with light-yellow discoloration and swelling, which showed obvious fatty liver characteristics, while the livers of the NC group demonstrated normal morphology in terms of size, shape, texture, and color (bright reddish-brown); however, after treatment with Simvastatin and CGE, the hepatic steatosis was reduced to different degrees in comparison to the MOD group. As a result, it was found that CGE can inhibit hepatic lipogenesis and fat deposition.

H&E staining showed that hepatic lobule architecture in the NC group was clear, with normal structure, no visible lipid droplets, and fat vacuoles. In contrast, the hepatic lobules of MOD group were destroyed with cell chaos, increased lipid droplets and necrosis, and there were visible fat vacuoles in the liver cells, while the lipid droplets and fat vacuoles in the hepatic lobules were obviously reduced after treatment with Simvastatin and CGE, and the hepatocytes in liver steatosis were improved to varying degrees, particularly CGE-H. Detailed results are illustrated in Figure 5B.

ORO staining showed that the NC group had sparse ORO-stained lipid droplet distribution throughout the liver tissue. Conversely, there was a great number of ORO-stained lipid droplets accumulated in the hepatocytes of MOD group. However, fewer red-stained lipid droplets of hepatocytes were observed after treatment with CGE and Simvastatin (Figure 5C).

Masson stain revealed that there were increased collagen fibers in the hepatic tissues of the MOD group, while the liver tissues of the NC group had fewer collagen fibers. Compared with the MOD group, the collagen fibers throughout the hepatic tissue decreased homogeneously after treatment with CGE and Simvastatin (Figure 5D).

### 3.6. Protein Expression Levels of TGF-β, PI3K, PPAR-γ, AKT, LXR-α, and ABCA1 in Aortas and Livers of ApoE^−/−^ Mice

TGF-β/PI3K/AKT signaling pathway was correlated to lipid metabolism, vascular endothelial dysfunction, and inflammation [19,31]. PPARγ exerts anti-atherosclerosis effects by promoting the removal of cholesterol from macrophages via cholesterol transporters such as ABCA1 protein [32]. The study results are shown in Figure 6. In the NC group, the protein expression of PPAR-γ, LXR-α, ABCA1, and TGF-β was high (Figure 6A,E), while that of PI3K and AKT1 was low in the aorta (Figure 6A). In the MOD group, the protein expression of PPAR-γ, TGF-β, ABCA1, and LXR-α down-regulated (*p* < 0.05, *p* < 0.01, *p* < 0.001) (Figure 6B,F–H), while those of PI3K and AKT1 increased (*p* < 0.05, *p* < 0.01, *p* < 0.001) in the aorta when compared with the NC group (Figure 6C,D). CGE and Simvastatin upregulated the expression levels of LXR-α, ABCA1, PPAR-γ, and TGF-β proteins (Figure 6A,E) and downregulated the expression levels of PI3K and AKT1 proteins (Figure 6A) when compared with the MOD group.

In addition, the mRNA expressions of TGF-β, PI3K, ABCA1, LXR-α, PPAR-γ, and AKT1 in mouse livers were assayed with q-PCR. As shown in Figure 7, the TGF-β, LXR-α, PPAR-γ, and ABCA1 expression levels of the livers in the MOD group decreased significantly in comparison with the NC group (*p* < 0.01) (Figure 7A,D–F). In contrast, the mRNA expressions of TGF-β, PPAR-γ, LXR-α and ABCA1 after treatment with Simvastatin and CGE were significantly upregulated in comparison with the MOD group (*p* < 0.01) (Figure 7A,D–F), and this upregulation effect of CGE is dose-dependent. In addition, PI3K and AKT1 mRNA expression levels in the livers of the MOD group were significantly increased in comparison with the NC group, while marked decreases in PI3K and Akt1 mRNA expressions after administration of the Simvastatin and CGE were observed (*p* < 0.05, *p* < 0.01) (Figure 7B,C). In the same way, the regulatory effects of CGE on PI3K and AKT1 mRNA also showed a dose-dependent relationship.

As shown in Figure 7, a significant increase in TGF-β, LXR-α, PPAR-γ, and ABCA1 protein was observed in the livers of apoE^−/−^ mice treated with CGE when compared with the model group (Figure 7G,K). In addition, we also observed a marked downregulation of PI3K and Akt1 protein expression after treatment with Simvastatin and CGE, while the expression levels of PI3K and Akt1 proteins were upregulated in the model group (Figure 7I,J), with the regulation effects of CGE in demonstrating a dose-dependent relationship. Western Blot analysis revealed that Simvastatin and CGE can significantly upregulate the expression levels of TGF-β, LXR-α, PPAR-γ, and ABCA1 protein (Figure 7H,L–N) and downregulate the protein expression of PI3K and Akt1 in the livers of the model group. The results matched the changes in these protein expression levels in the aorta very well.

## 4. Discussion

AS is by far the most frequent underlying cause of carotid artery disease, coronary artery disease, and peripheral arterial disease [18], and is responsible for almost all obstructive processes in the arteries [33], which, in turn, are the dominant causes of cardiovascular diseases (CVDs). A total of 17.9 million lives are estimated to be lost to CVDs each year [34]; hence, atherosclerotic CVDs are understood to be the leading cause of death worldwide [35]. In addition, AS is considered to be a lifestyle-related disease because its incidence is closely associated with life habits, and the role of cholesterol in the development of AS has been demonstrated [36].

In the literature, the pathogenesis of AS has been suggested to be a chronic inflammatory disease caused by the interactions of atherogenic lipoproteins and vascular wall cells, including monocyte/macrophages, mast cells, endothelial cells, lymphocytes, and smooth muscle cells [37]. Among them, macrophage activation or dysfunction may play a major role in the initiation and progression of ASCVD, which is attributed to lipoprotein ingestion and accumulation by arterial macrophages, which produce foam cells [38]. Macrophages are derived from multiple differentiations. The adhesion of blood monocytes to endothelial cells is the earliest event in the arterial intima; following this, the monocytes emigrate into the sub-intima, where they differentiate into macrophages [39]. In addition, evidence suggests that vascular smooth muscle cells can dedifferentiate to a macrophage-like plaque state. The progression of ASCVD is mainly derived from local macrophage proliferation [40].

Macrophages commonly have two distinct subsets (M1 and M2), which have unique abilities in terms of their repairing inflammation-associated injury or destroying pathogens, and the M1/M2 polarization balance governs the fate of an organ (i.e., inflammation or injury). M2 macrophages secrete great amounts of TGF-β and IL-10 to suppress inflammation, contributing to vasculogenesis, tissue repair, remodeling, and maintaining homeostasis [41]. In addition, the PI3K/AKT1 pathway in macrophages plays distinct roles in macrophage biology and inflammatory disease regulation [42] and may impact the development of AS. The suppression of the PI3K/ AKT1 signaling pathway will markedly reduce the proliferation and viability of blood monocytes and macrophages with the suppression of AS [20]. As a result, the interaction or synergistic effects between TGF-β and PI3K/AKT1 pathways regulate cell proliferation and thereby maintain tissue homeostasis.

Of the genetically engineered models, apoE- or Ldlr-deficient hyperlipidemic mice are the two most commonly used mouse models for AS research. Of the two, the apoE^−/−^ model is the only one that demonstrated extensive atherosclerotic changes on a chow diet [43,44]. In this study, the atherosclerosis models of lipid deposition at the whole aorta were constructed successfully using apoE^−/−^ mice fed with an HFD.

To our knowledge, TGF-β, as a cytokine, is most active in lipid-rich aortic intimal lesions [45]. Studies have suggested that TGF-β has bidirectional roles in AS. In the early stage of AS, it has antioxidant and anti-inflammatory effects, while in the late stage of AS, TGF-β can facilitate vascular cell proliferation and accelerate plaque formation [31,46]. In recent years, studies have found that TGF-β plays a major role in the stability of AS plaques, the inhibition of monocyte/macrophage chemotaxis, the absorption of oxidized LDL, and the reduction in VSMC proliferation and lipid absorption, as well as being anti-inflammatory [47]. PI3K/AKT1 is an important downstream pathway of TGF-β. It is possible that systemic administration of PI3K/AKT1 inhibitors may be correlated with increased apoptosis in macrophages and, during the early stages of AS, may suppress lesion progression [48]. In this study, concerning HFD-induced hyperlipidemia and AS in apoE^−/−^ mice, our findings indicate that the blood lipids were abnormal and aortic plaque and liver fat lesions were obvious (Figure 2, Figure 3 and Figure 4). Moreover, the protein and mRNA expression of TGF-β in the aorta and livers were significantly reduced, while those of PI3K and AKT1 increased (*p* < 0.05, *p* < 0.01) (Figure 5 and Figure 6). After 16 weeks of treatment with CGE, the protein expression of TGF-β increased significantly. At the same time, that of PI3K and AKT1 decreased, and the dyslipidemia, aortic intima, and liver lesions improved to a certain extent, suggesting that CGE can regulate vascular wall cells by regulating the TGF-β/PI3K/AKT1 signaling pathways associated with tissue homeostasis. Their activation can have a negative effect on the interactions between atherogenic lipoproteins and vascular wall cells.

In the atherosclerotic and hepatic steatosis in mice induced by hyperlipidemia, the cholesterol efflux caused by hyperlipidemia is restored by regulating the LXR-α/β-PPAR-γ pathway [49]. As lipid sensors that regulate glucose and lipid metabolism, PPAR-γ and LXR-α are activated by oxysterols and oxidized fatty acid ligands, respectively. These intracellular cholesterol sensors induce ABCG1 and ABCA1 expression and remove excess cholesterol [50,51]. Meanwhile, current studies have revealed that the activation of PPAR-γ can reduce the inflammatory factors in the aortic root, thus inhibiting the initiation and progression of AS [32]. In this study, based on the atherosclerosis apoE^−/−^ mice model induced by an HFD, we further verified the effect of CGE on the PPAR-γ-LXR-α-ABCA1 signaling pathway. As a result, in both the liver and aorta tissues of HFD-fed apoE^−/−^ mice, CGE upregulated the protein expression levels of LXR-α, PPAR-γ, and ABCA1, as well as the mRNA expression levels, which demonstrated an obvious difference in comparison with the MOD group (*p* < 0.05, *p* < 0.01) (Figure 5 and Figure 6). Accordingly, for the atherosclerosis apoE^−/−^ mice model, CGE decreased their lipid levels and improved liver lipid deposition, liver lesions, and aortic plaques, suggesting that CGE can regulate lipid metabolism and cholesterol synthesis, as well as lipid transportation by upregulating the PPAR-γ-LXR-α-ABCA1 signaling pathways associated with reverse cholesterol transport. Their activation could lead to the removal of excessive intracellular cholesterol and a reduction in the formation of foam cells, which would have a negative effect on de novo atherogenesis.

## 5. Conclusions

CGE can mitigate high-fat-diet-induced high circulating lipid levels, liver lipid deposition, and aorta lipid accumulation by regulating abnormal lipid metabolism and cholesterol synthesis while reducing the formation of atherosclerotic plaques and foam cells by inhibiting the interactions of atherogenic lipoproteins (ox-LDL) with vascular wall cells. These effects of CGE are likely correlated to the TGF-β/PI3K/ALT1 and PPARγ/LXRα/ABCA1 signaling pathways.

CGE, which is helpful in the prevention of abnormal lipid metabolism and atherosclerosis, can be used as a healthcare dietary supplement. But the active components associated with the beneficial effects highlighted above require further study.

## Figures and Tables

**Figure 1 foods-14-04267-f001:**
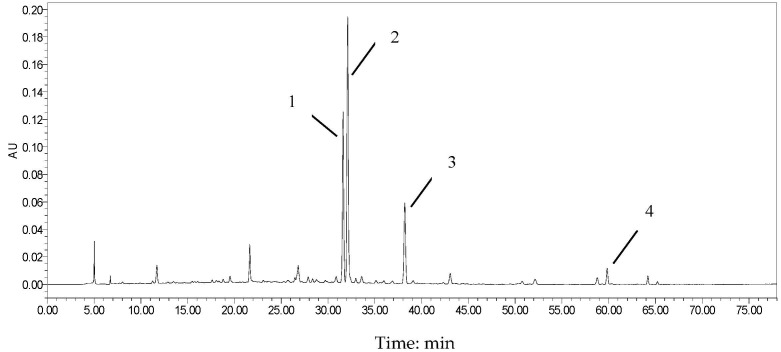
HPLC fingerprint of CGE (Peak 1, 2, 3, and 4 is rhoifolin, naringin, meranzin hydrate, and isomerazin, respectively).

**Figure 2 foods-14-04267-f002:**
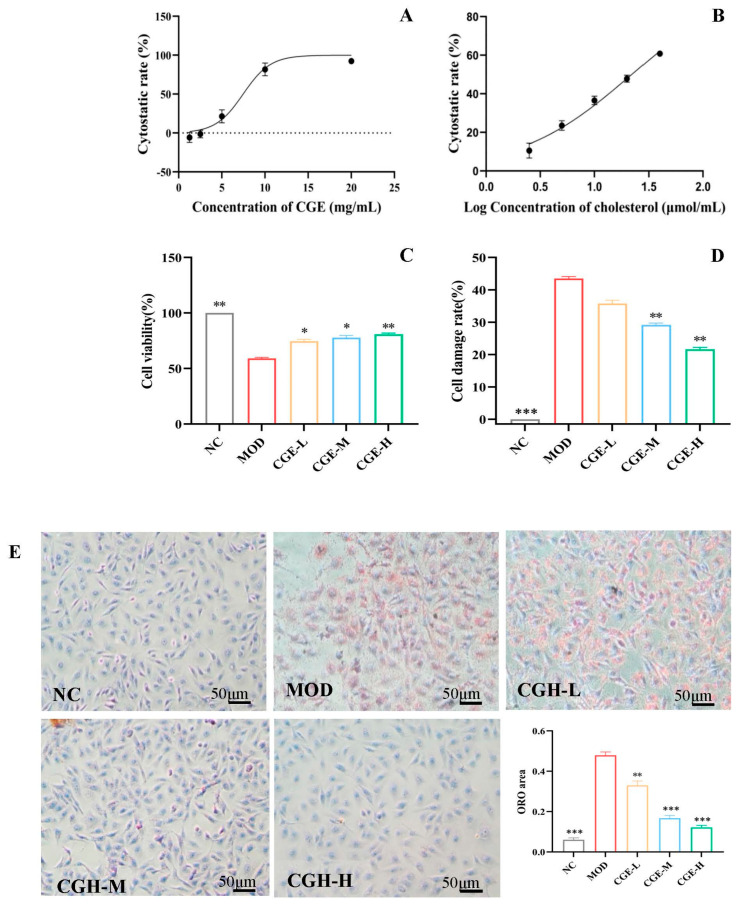
The protective effects of CGE on cholesterol-induced HUVECs injury (Appendix A). (**A**) Dose–inhibition curve of CGE on HUVECs’ viability. (**B**) The dose–cytostatic relationship curve of cholesterol on HUVECs cells. (**C**) The protective effects of CGE on cholesterol (10 μM)-induced cell injury (Partial η^2^ = 0.961). (**D**) The dose–response relationship of CGE on LDH release levels in the absence or presence of 10 μM cholesterol (Partial η^2^ = 0.993). (**E**) Lipid accumulation of HUVECs cultured with CGE for 24 in the presence or absence of 10 μΜ cholesterol (Partial η^2^ = 0.973). Intracellular lipid droplets stained with ORO were visualized using light microscopy (200×; Scale bar = 50 μm). Representative images of the normal control group (NC), HUVECs treated with 10 μM cholesterol for 24 h (MOD), and model HUVECs treated with CGE at 1.25, 2.50, and 5.00 mg/mL (final concentration in the well), respectively, for 24 h, which were stained with ORO. (**F**) Anti-migration effects of CGE on ox-LDL-induced HUVECs (×100, Scale bar = 200 μm). (**G**) Anti-invasion effects of CGE on HUVECs. HUVECs were treated with ox-LDL (50 µg/mL) and CGE (1.25, 2.50, 5.00 mg/mL), and transwell was used for the invasion assays in HUVECs. The pictures of the positive cells stained with 0.1% crystal violet solution were photographed using a microscope (×200, Scale bar = 50 μm). The number of positive cells was analyzed using the Image J software (version: 1.54j, NIH, Bethesda, MD, USA), and data were represented as $\bar{x}$ ± SE (*n* = 3), *p* < 0.001 vs. MOD cells (HUVECs treated by ox-LDL alone). (**H**) Quantitative analysis of the migration area in (**F**) (*n* = 6) (Wilks’ Λ = 0.003, R^2^ = 0.986, Eta^2^ = 0.986). (**I**) Quantitative analysis of the migration cells in (**G**) (*n* = 6) (Wilks’ Λ = 0.003, R^2^ = 0.986, Eta^2^ = 0.986). The error bars represent $\bar{x}$ ± SE. Difference was considered statistically significant when * *p* < 0.05, ** *p* < 0.01, and *** *p* < 0.001 vs. MOD cells; *n* = 6 for each group.

**Figure 3 foods-14-04267-f003:**
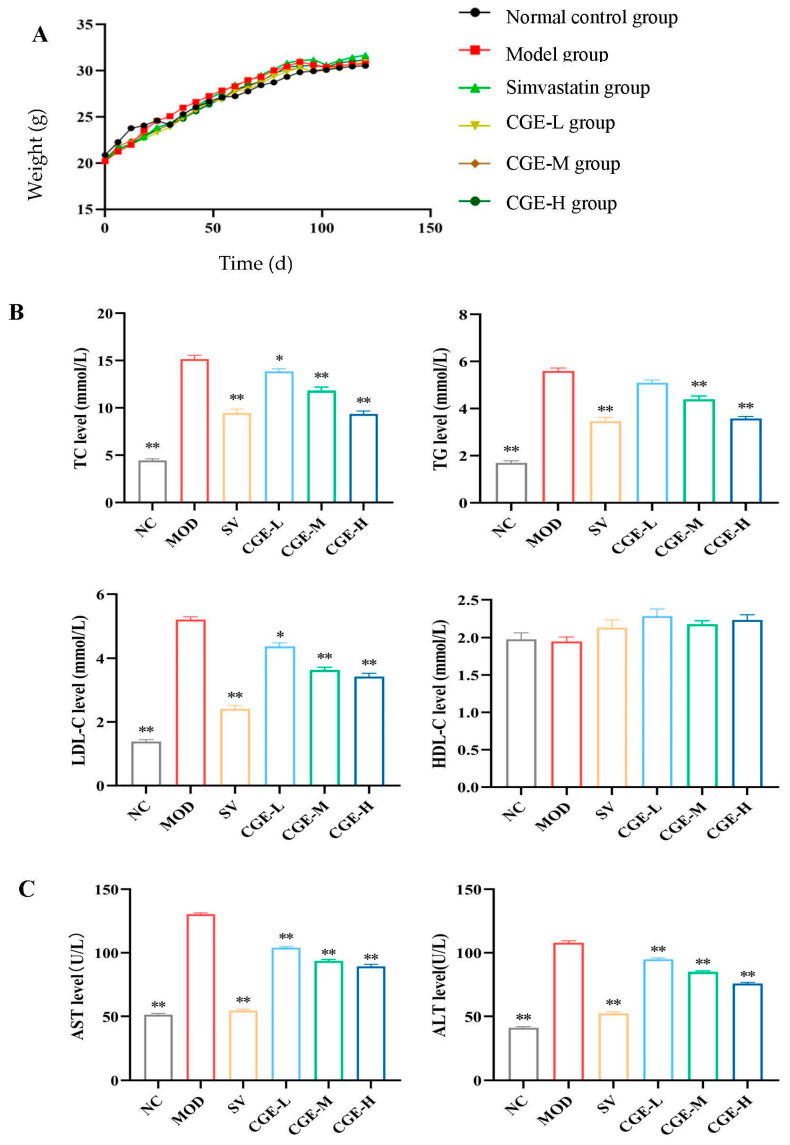
Effects of CGE on body weight and serum lipid profile (Appendix A). (**A**) The dynamic body weights of apoE^−/−^ mice. (**B**) Comparative analysis of serum lipid levels in each group. The levels of HDL-C, TC, LDL-C, and TG are shown in mmol/L (Wilks’ Λ = 0.002, R^2^ = 0.963, Eta^2^ = 0.977). (**C**) Comparison of serum ALT and AST (Partial η^2^ =0.992). The activities of ALT and AST are shown in U/L. The error bars represented $\bar{x}$ ± SE (*n* = 10). It indicates a statistically significant difference when ** *p* < 0.01 and * *p* < 0.05 vs. the model control group.

**Figure 4 foods-14-04267-f004:**
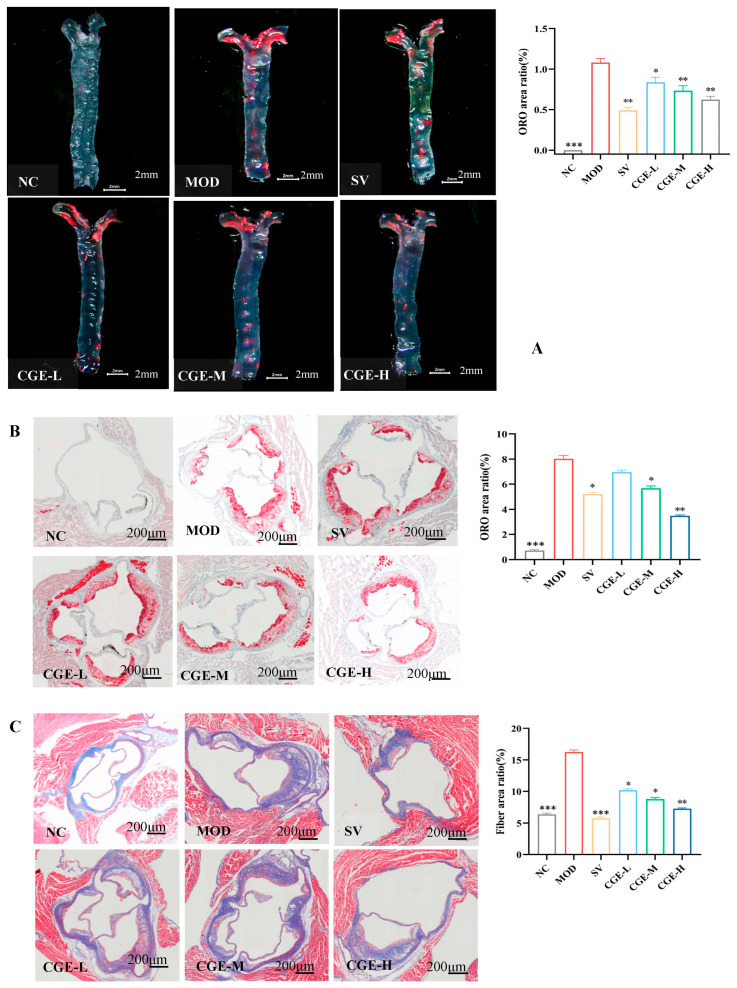
Pathological changes observation regarding mouse aortas (Appendix A). (**A**) Representative ORO staining images of an en face aorta (Partial η^2^ = 0.961). Light-field microscopy images of the positive area in aorta intima; scale bar = 2 mm. (**B**) Lipid accumulation of the cross-section of the aortic roots (Partial η^2^ = 0.982) (ORO staining; scale bar = 200 μm). (**C**) The area of plaque collagen fiber of the aortic roots (Partial η^2^ = 0.983) (Masson stain; scale bar = 200 μm). (**D**) Representative HE staining images of aortic cross-sections (HE stain; scale bar = 200 μm). (**E**) Representative ultrastructural images of an aortic cross-section (voltage 80 KV, 3000×). Scale bar = 2 μm. NC: normal C57BL mice (with normal diet); MOD: model group; SIV: Simvastatin group; CGE-L, CGE-M, and CGE-H: HFD-fed apoE^−/−^ mice treated with CGE at 1.25, 2.50, and 5.00 mg/mL, respectively, for 16 weeks. The error bars represent $\bar{x}$ ± SE. Difference was considered statistically significant when * *p* < 0.05, ** *p* < 0.01, and *** *p* < 0.001 vs. MOD group.

**Figure 5 foods-14-04267-f005:**
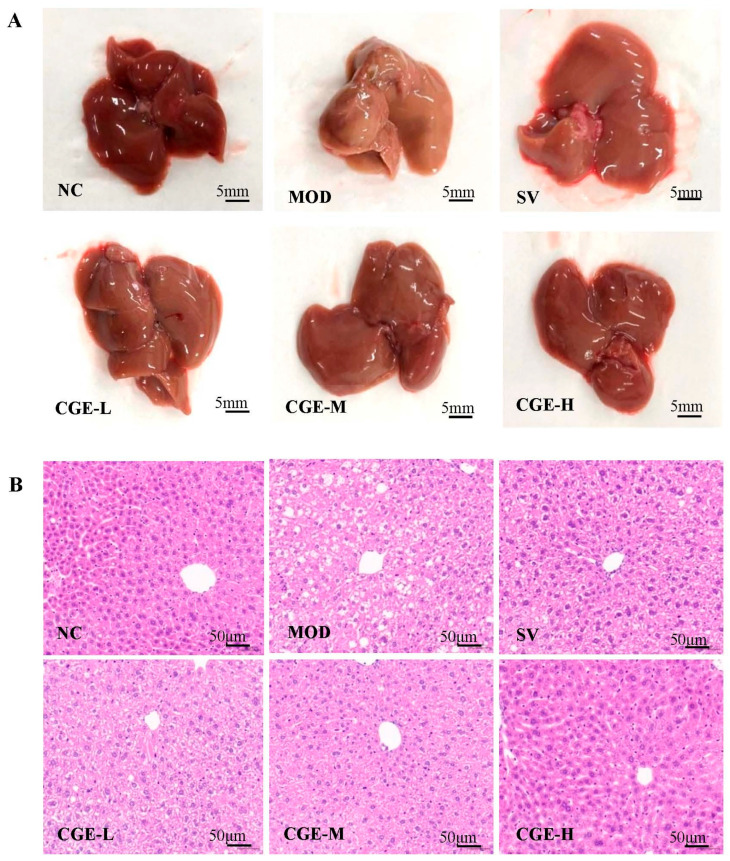
Effect of CGE on histopathological changes in the livers (Appendix A). (**A**) The gross morphology observation of mouse livers (scale bar = 5 mm). (**B**) Representative HE staining images of mouse livers (HE stain, scale bar = 50 μm). (**C**) Representative oil red O staining images of mouse livers (Partial η^2^ = 0.989) (ORO stain, scale bar = 200 μm). (**D**) The area of the collagen fiber of mouse livers (Partial η^2^ = 0.994) (Masson stain, scale bar = 200 μm). The error bars represented $\bar{x}$ ± SE (*n* = 5). Difference was considered statistically significant when * *p* < 0.05, ** *p* < 0.01 and *** *p* < 0.001 vs. the model control group.

**Figure 6 foods-14-04267-f006:**
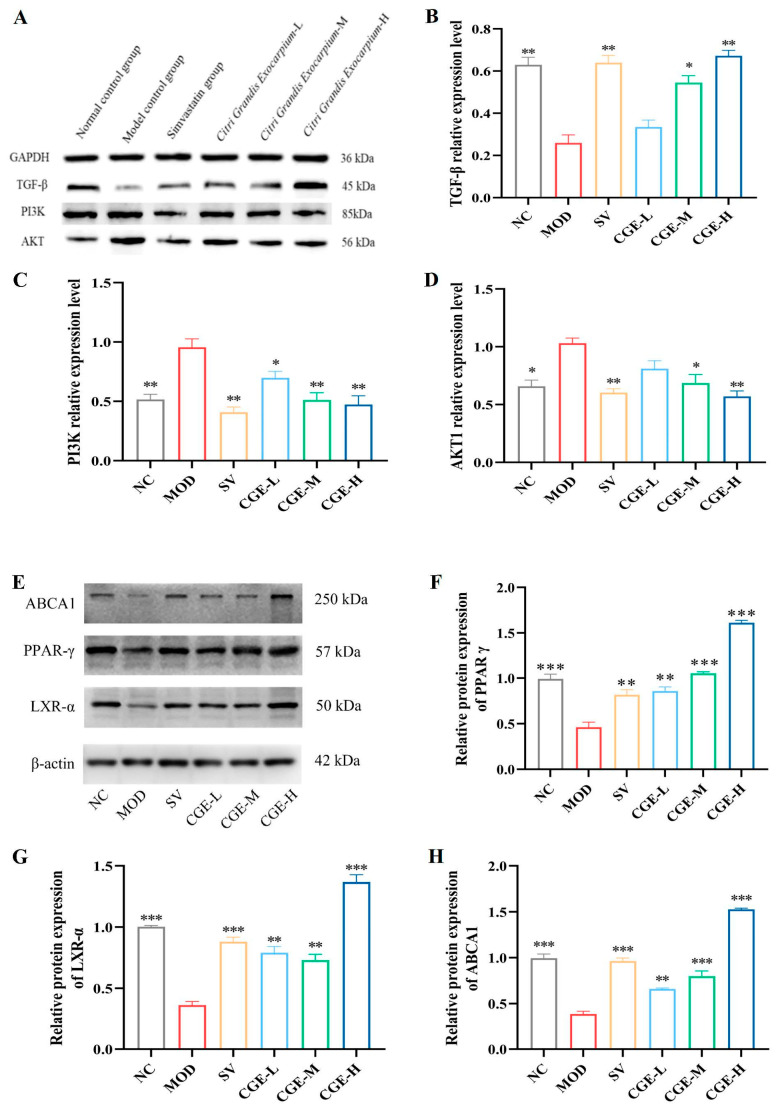
Representative blotting images and protein densitometry showing the relative expression levels of TGF-β, PI3K, AKT1, PPAR-γ, LXR-α, and ABCA1 proteins normalized with GAPDH and β-actin in aorta of apoE^−/−^ mice (Appendix A). (**A**,**E**) are the representative protein blotting images; (**B**–**D**,**F**–**H**) Expression levels of TGF-β, PI3K, AKT1, PPAR-γ, LXR-α, and ABCA1 proteins in aorta were shown in the bar chart and the blotting grayscale value analyses normalized with GAPDH and β-actin (Wilks’ Λ = 0.003, R^2^ = 0.970, Eta^2^ = 0.970), which were used as loading control. The error bars represented $\bar{x}$ ± SE. * *p* < 0.05, ** *p* < 0.01, and *** *p* < 0.001 vs. the MOD group indicate a statistically significant difference; *n* = at least 3 for each group.

**Figure 7 foods-14-04267-f007:**
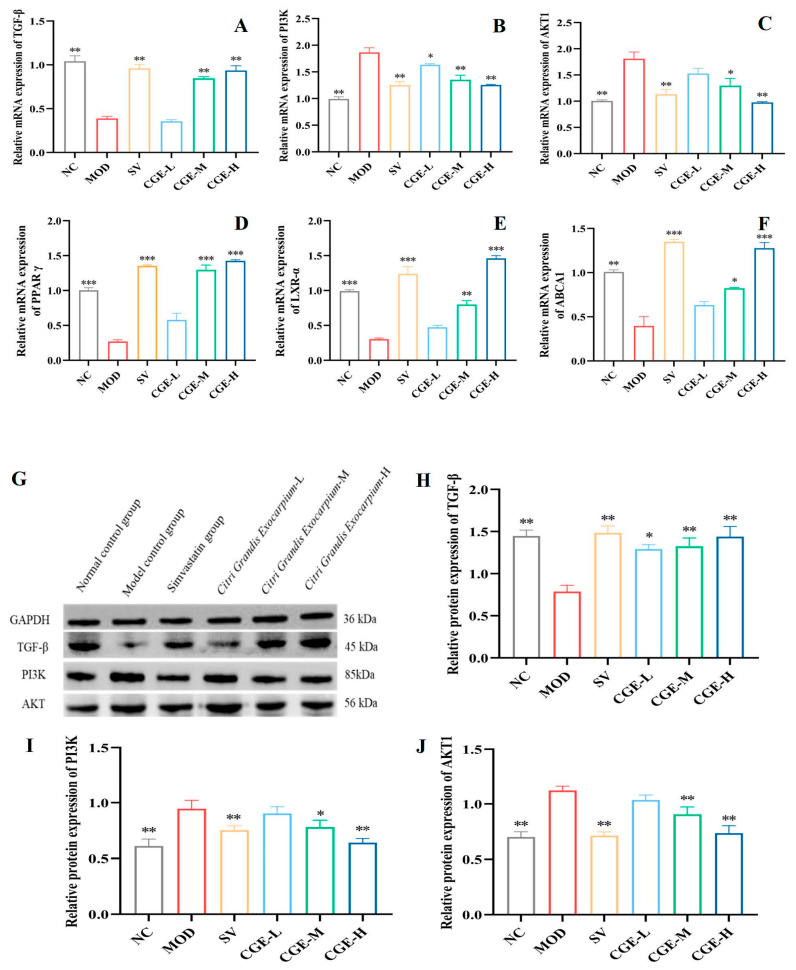
Effect of CGE on the expression of TGF-β, PI3K, AKT, PPAR-γ, LXR-α, and ABCA1 in mouse livers (Appendix A). (**A**–**F**) represent the mRNA expression levels of TGF-β, PI3K, AKT1, PPAR-γ, LXR-α, and ABCA1 to GAPDH in livers, respectively (Wilks’ Λ = 0.001, R^2^ = 0.953, Eta^2^ = 0.958). (**G**,**K**) are the representative protein blotting images; (**H**–**J**,**L**–**N**) are the respective optical density ratios of bands, which represent the relative expression levels of TGF-β, PI3K, AKT, PPAR-γ, LXR-α, and ABCA1 proteins normalized with GAPDH or β-actin in the livers of apoE^−/−^ mice (Wilks’ Λ = 0.008, R^2^ = 0.986, Eta^2^ = 0.986). The blotting grayscale value was analyzed relative to the GAPDH band and β-actin, as a loading control. The error bars represent the $\bar{x}$ ± SE. It is considered to have statistically significant differences when * *p* < 0.05, ** *p* < 0.01, and *** *p* < 0.001 vs. the MOD group; *n* = at least 3 for each group.

**Table 1 foods-14-04267-t001:** Primer sequences of target genes and reference genes in mice.

Primer Name	Primer Sequence (5′–3′)	Product Length (bp)
TGF-β1-F	TGGCCAGATCCTGTCCAAAC	97
TGF-β1-R	GTTGTACAAAGCGAGCACCG	97
PI3K-F	GGGACTTTGGAGACCAGAAGG	108
PI3K-R	GTGTCTGGGTTCACCACACC	108
AKT1- F	AGTCCCCACTCAACAACTTCT	119
AKT1-R	GAAGGTGCGCTCAATGACTG	119
PPAR-γ-F	GCAGCTACTGCATGTGATCAAGA	107
PPAR-γ-R	GTCAGCGGGTGGGACTTTC	107
ABCA1-F	GAGCAAAGCCAAGCATCTTC	172
ABCA1-R	TAGAACGGGCAGGTTGGTAG	172
LXR-α-F	AGGAGTGTCGACTTCGCAAA	101
LXR-α-R	CTCTTCTTGCCGCTTCAGTTT	101
GAPDH-F	AGGTCGGTGTGAACGGATTTG	123
GAPDH-R	TGTAGACCATGTAGTTGAGGTCA	123
β-actin-F	GGCTGTATTCCCCTCCATCG	154
β-actin-R	CCAGTTGGTAACAATGCCATGT	154

## Data Availability

The original contributions presented in the study are included in the article/Appendix A; further inquiries can be directed to the corresponding author.

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
