# Peer review of "Citri grandis Exocarpium Extract Alleviates Atherosclerosis in ApoE−/− Mice by Modulating the Expression of TGF-β1, PI3K, AKT1, PPAR-γ, LXR-α, and ABCA1"

_foods, 2025, doi:10.3390/foods14244267_

Round 1
Reviewer 1 Report
Comments and Suggestions for Authors
Dear Authors,
The study is very interesting and contains results that will be useful for this field of knowledge. However, before publication, several points related to reproducibility and repeatability must be adjusted and corrected. The section I specifically reviewed is on statistical analyses, and this is evidently reflected in the results.
1. The mean values should be graphed with standard error bars, because inferential parametric tests were applied (see Cumming, G., Fidler, F., & Vaux, D. L. (2007). Error bars in experimental biology. The Journal of Cell Biology, 177(1), 7-11).
2. These tests (one-way ANOVA and independent samples t-test) require three assumptions: data independence, compliance with the normal error distribution, and homogeneity of variances; the latter indicates that they were verified. However, they must indicate what type of hypothesis test was applied. Likewise, with the normal error distribution, indicate the test and statistic values.
3. They applied more than 60 one-way ANOVAs, which inflates the probability of committing type II statistical errors. In this sense, it is advisable to perform MANOVAs with a matrix of 18 rows (n=18) by 18 columns (response variables at most), so as not to violate the assumption of symmetry of the data matrix. In this sense, the minimum number of response variables must be 5 for the MANOVA. Therefore, they must readjust their analyses.
4. Applying this type of analysis requires not violating the assumptions of normal error distribution and homogeneity of variances. Therefore, there are several options: transform the data to the base 10 logarithm, or rank (Conover, W. J., & Iman, R. L. (1981). Rank transformations as a bridge between parametric and nonparametric statistics. The American Statistician, 35(3), 124-129).
Reanalysis is essential to minimize type II statistical errors. It is common practice to apply too many independent tests, and this deficiency can be resolved with MANOVA.
Best regards,
Comments on the Quality of English Language
The English could be improved to more.
Author Response
The study is very interesting and contains results that will be useful for this field of knowledge.
Response: We really appreciate the reviewer’s kind comments.
However, before publication, several points related to reproducibility and repeatability must be adjusted and corrected. The section I specifically reviewed is on statistical analyses, and this is evidently reflected in the results.
Review Comments and Suggestions for Author:
- The mean values should be graphed with standard error bars, because inferential parametric tests were applied (see Cumming, G., Fidler, F., & Vaux, D. L. (2007). Error bars in experimental biology. The Journal of Cell Biology, 177(1), 7-11).
- These tests (one-way ANOVA and independent samples t-test) require three assumptions: data independence, compliance with the normal error distribution, and homogeneity of variances; the latter indicates that they were verified. However, they must indicate what type of hypothesis test was applied. Likewise, with the normal error distribution, indicate the test and statistic values.
- They applied more than 60 one-way ANOVAs, which inflates the probability of committing type II statistical errors. In this sense, it is advisable to perform MANOVAs with a matrix of 18 rows (n=18) by 18 columns (response variables at most), so as not to violate the assumption of symmetry of the data matrix. In this sense, the minimum number of response variables must be 5 for the MANOVA. Therefore, they must readjust their analyses.
- Applying this type of analysis requires not violating the assumptions of normal error distribution and homogeneity of variances. Therefore, there are several options: transform the data to the base 10 logarithm, or rank (Conover, W. J., & Iman, R. L. (1981). Rank transformations as a bridge between parametric and nonparametric statistics. The American Statistician, 35(3), 124-129).
Reanalysis is essential to minimize type II statistical errors. It is common practice to apply too many independent tests, and this deficiency can be resolved with MANOVA.
Best regards,
Response:
You raised excellent comments. These opinions and suggestions are all valuable and very helpful for improving our paper, as well as the important guiding role to our manuscript. According to the suggested, the statistical results are presented as the mean ± standard error (SE) in all figures, and the mean values were graphed with standard error bars.
In addition, the comparison of the measured variables between groups was performed using Multi-factor ANOVA method combined with Shapiro-Wilk and Wilks’ Lambda and so on, and the corrections were shown in Fig.2, Fig.3, Fig.4, Fig.5 and Fig. 6 in the revised manuscript.
Comments on the Quality of English Language
The English could be improved to more.
Response:
Thank you for the kind reminds. In the revised manuscript, we carefully carry out modifications.
Reviewer 2 Report
Comments and Suggestions for Authors
Dear Authors,
I would like to thank you for preparing this engaging manuscript. I have some comments that could enhance its scientific quality.
First, the entire manuscript requires language editing. All abbreviations in the abstract should be spelled out.
Additionally, please include a description of the chemical composition and nutritional components of Citri Grandis exocarpium in the introduction.
In the Methods section, specifically under Cell Lines and Animals, include details on how the animal models are categorized into different experimental groups, specifying the number of groups and the number of mice in each group. Remember to cite the reference for the “Guide for Care and Use of Laboratory Animals.”
Ensure that all figures are numbered and titled; they should be self-explanatory. Include the full names of any abbreviations in the legends and indicate the number of subjects.
In the discussion, elaborate on the proposed mechanisms by which Citri Grandis exocarpium regulates abnormal lipid metabolism and cholesterol synthesis, as well as its role in reducing atherosclerotic plaque formation and foam cells. The beneficial effects may be linked to polyphenols, flavonoids, vitamins, and minerals that act as antioxidants or inhibitors of enzymes involved in cholesterol synthesis.
Finally, please address the strengths and limitations of the study at the end of the discussion.
Author Response
Review Comments and Suggestions for Author:
I would like to thank you for preparing this engaging manuscript. I have some comments that could enhance its scientific quality.
- First, the entire manuscript requires language editing. All abbreviations in the abstract should be spelled out.
Response:
That’s really a good suggestion. Thank you so much for reminding us. In the revised manuscript, we had implemented the modifications on the abbreviations in the abstract.
- Additionally, please include a description of the chemical composition and nutritional components of Citri Grandis exocarpium in the introduction.
Response:
Thanks a lot for the nice suggestion. A short description for the chemical composition and nutritional components of Citri Grandis exocarpium was added in the revised manuscript on Line 18-22, Page 2. The description was expressed as follows.
“…and as well as chemical composition (Wang LJ, et al., 2012), mainly including flavonoids (Deng SD, et al., 2013; Zou MH, et al., 2024), volatile oil (Su Z, et al., 2019; Xie ZS, et al., 2013), polysaccharide (Chen M, et al. 2025), coumarin (Xiao WQ, et al., 2009; Chen ZX, et al., 2004) and inositol (Chen DL, et al. 2012; Zhang XQ, et al. 2013), etc., in particular naringin, naringenin and rhoifolin, are the main active ingredients (Sun G, et al., 2010;ChP, 2020 Edition).
- In the Methods section, specifically under Cell Lines and Animals, include details on how the animal models are categorized into different experimental groups, specifying the number of groups and the number of mice in each group. Remember to cite the reference for the “Guide for Care and Use of Laboratory Animals.”
Response:
Thank you so much for reminding us. In the revised manuscript, we added the reference for the “Guide for Care and Use of Laboratory Animals” in the part 2.2 “Cell Lines and Animals” and part 2.10 “Modeling and grouping” of the Materials and Methods section.
- Ensure that all figures are numbered and titled; they should be self-explanatory. Include the full names of any abbreviations in the legends and indicate the number of subjects.
Response:
We sincerely apologize for our mistakes. The reviewer read so carefully about our manuscript and thanks for your time! We had carefully read the manuscript and made revisions.
- In the discussion, elaborate on the proposed mechanisms by which Citri Grandis exocarpium regulates abnormal lipid metabolism and cholesterol synthesis, as well as its role in reducing atherosclerotic plaque formation and foam cells. The beneficial effects may be linked to polyphenols, flavonoids, vitamins, and minerals that act as antioxidants or inhibitors of enzymes involved in cholesterol synthesis.
Finally, please address the strengths and limitations of the study at the end of the discussion.
Response:
What the reviewer said is right. It’s really true that there must be some chemical composition and nutritional components associated with these beneficial effects of Citri Grandis exocarpium extract (CGE). As suggested, we supplemented the limitations of the study at the end of the discussion in the revised manuscript. The description was expressed as follows: “But the active components associated with the beneficial effects needs to be further revealed.”
Anyway, the purpose of this study was to confirm the preventative effects of CGE in atherosclerosis, and tried to explore the effect mechanism. This work is very essential basis for further active components investigation, which opens up the window for it. Since the complicacy might not be resolved completely in one study.
Reviewer 3 Report
Comments and Suggestions for Authors
The manuscript addresses a topic in dyslipidemic management and presents interesting data. Overall, the study is well-organized, but a few points could be clarified or discussed further to strengthen the manuscript and improve its interpretability:
1. In the Keywords section, many terms are repeated from the title. Revising this could enhance the visibility and discoverability of the manuscript.
2. Please consider the use of the term hyperlipidemia, I believe that dyslipidemia is more correct
3. Part 2.3. Please explain how you determined the experimental conditions.
4. The fonts in the work itself are mirrored, uniformed.
5. Check all figures and tables. They must be mentioned in the text, adequately labeled. Some figures do not have good x- and y-axis labels. One figure also has labels in Chinese.
6. The results and discussion are nicely written.
Author Response
Comments and Suggestions for Authors:
The manuscript addresses a topic in dyslipidemic management and presents interesting data. Overall, the study is well-organized, but a few points could be clarified or discussed further to strengthen the manuscript and improve its interpretability:
- In the Keywords section, many terms are repeated from the title. Revising this could enhance the visibility and discoverability of the manuscript.
Response: Thanks for the good suggestion. In the revised manuscript, the Keywords has been replaced with “Citri Grandis exocarpium; Atherosclerosis; Lipid metabolism regulation; Reverse cholesterol transport”.
- Please consider the use of the term hyperlipidemia, I believe that dyslipidemia is more correct.
Response:
That’s really a nice question. We really admire the review’s stringent academic spirit. According to the suggested, we looked up some academic literatures. The reported review articles suggested that disorder in lipoprotein metabolism (dyslipidemia) can be classified as hypercholesterolemia, hypertriglyceridemia, combined hyperlipidemia, and low levels of high density [1]. Hyperlipidemia, the most common form of dyslipidemia, is the main source of cardiovascular disorders, characterized by elevated level of total cholesterol (TC), triglycerides (TG) and low-density lipoprotein cholesterol (LDL-C) with high-density lipoprotein cholesterol (HDL-C) in peripheral blood [2]. In professional literature, hyperlipidemia emphasizes the objective indicators of "blood lipid abnormalities" (such as excessive values of total cholesterol, triglycerides, LDL-C etc.), while dyslipidemia focuses on the broad concept of "blood lipid abnormalities", which includes both numerical abnormalities and lipoprotein structural abnormalities. For example, Hyperlipidemia: Specific numerical abnormalities such as total cholesterol ≥ 5.2mmol/L or triglycerides ≥ 1.7mmol/L. Dyslipidemia: a comprehensive assessment covering indicators such as total cholesterol, LDL-C, HDL-C. However, our study demonstrated that CGE could regulate the numerical abnormalities of hyperlipidemia, but it has still not been well unclear whether modulate lipoprotein structural abnormalities or not.
Hyperlipidemia and dyslipidemia are different expressions of the same concept in medicine, with no essential difference [3, 4, 5]. But hyperlipidemia is more commonly used in international medical exchanges.
Based on the afore-mentioned reasons, we use the term hyperlipidemia in the manuscript.
Reference:
- I Jialal. A practical approach to the laboratory diagnosis of dyslipidemia. Am J Clin Pathol. 1996, 106(1):128-38. doi:10.1093/ajcp/106.1.128.
- Rauf A, Akram M, Anwar H, Daniyal M, Munir N, Bawazeer S, Bawazeer S, Rebezov M, Bouyahya A, Shariati MA, Thiruvengadam M, Sarsembenova O, Mabkhot YN, Islam MN, Emran TB, Hodak S, Zengin G, Khan H. Therapeutic potential of herbal medicine for the management of hyperlipidemia: latest updates. Environ Sci Pollut Res Int. 2022, 29(27): 40281-40301. doi: 10.1007/s11356-022-19733-7.
- Kreisberg RA, Oberman A. Medical management of hyperlipidemia/dyslipidemia. J Clin Endocrinol Metab. 2003, 88(6):2445-61. doi: 10.1210/jc.2003-030388.
- Stewart J, McCallin T, Martinez J, Chacko S, Yusuf S. Hyperlipidemia. Pediatr Rev. 2020, 41(8):393-402. doi: 10.1542/pir.2019-0053.
- Khalil YA, Rabès JP, Boileau C, Varret M. APOE gene variants in primary dyslipidemia. 2021, 328:11-22. doi: 10.1016/j.atherosclerosis.2021.05.007.
- Part 2.3. Please explain how you determined the experimental conditions.
Response:
That’s really a nice question. Actually, the preparation method of Citri Grandis exocarpium is closely correlated with the content of its components. In our previous research, with mass fractions of naringin and try extract yield as the evaluation indexes, the optimal conditions of Citri Grandis exocarpium were determined by using single factor test and orthogonal experimental design [1], the results showed that the optimal preparation conditions were as follows: extraction with filtered water, material-liquid ratio 1:30, extraction temperature 70℃, three times, 1hour for each time. The study on optimization of the preparation of CGE was published by us in Journal of Modern Food Science and Technology in June, 2024. In addition, the compositions of CGE were analyzed by HPLC chromatography (Fig.2). Hence, in this study, Part 2.3, the experimental conditions was determined based upon previous research on optimum extraction process of Citri Grandis exocarpium.
Reference
- LI Jinkun, JIN Gangliang, WANG Yanhui, XIAN Shaohua, ZHAO Junxin, YU Junlang, LIU Haofan, TAN Yaokang, WEN Junwei, XIONG Ping*. Optimization of the Preparation of Exocarpium Citri grandis Extract and Its Lipid Level-lowering Effects. Modern Food Science and Technology (Article in China). 2024, 40 (10):250-258.
- The fonts in the work itself are mirrored, uniformed.
Response:
Thank you so much for reminding us. The reviewer read so carefully about our manuscript and thanks for your time! We had carefully read the manuscript and made revisions.
- Check all figures and tables. They must be mentioned in the text, adequately labeled. Some figures do not have good x- and y-axis labels. One figure also has labels in Chinese.
Response:
Thank you so much for reminding us. We are really sorry for our carelessness and appreciated so much that the reviewer pointed it out. We did carry out modification in the revised manuscript.
- The results and discussion are nicely written.
Response: We really appreciate the reviewer’s kind comments.
Reviewer 4 Report
Comments and Suggestions for Authors
The submitted manuscript has scientific merit. I am requesting major revision for the following reasons.
1- The manuscript needs through English proofreading by a native English speaker. Almost every sentence has mistakes/ errors. The writing style prevents an honest evaluation of the manuscript. Below are some examples of language errors, and these are only from the abstract.
It was commonly consumed as a health drinks= remove “a” = as health drinks or remove”s” drinks=drink
However, more 11 about its nutritional and health care function was still not well understood.= this is not standard English= rewrite
can significantly decreased= decrease
HUVECs cells= either HUVECs or HUVEC cells.
oral administered= orally admisntered
LXR-α and ABCA1 in aortic and liver of= in aortic and liver tissues of
HUVECs cell stain= strain
2- Provide a clean version of the manuscript in which the changes are highlighted in green or yellow.
Author Response
Comments and Suggestions for Authors
The submitted manuscript has scientific merit. I am requesting major revision for the following reasons.
1- The manuscript needs through English proofreading by a native English speaker. Almost every sentence has mistakes / errors. The writing style prevents an honest evaluation of the manuscript. Below are some examples of language errors, and these are only from the abstract.
It was commonly consumed as a health drinks= remove “a” = as health drinks or remove”s” drinks=drink
However, more about its nutritional and health care function was still not well understood. = this is not standard English= rewrite
can significantly decreased= decrease
HUVECs cells= either HUVECs or HUVEC cells.
oral administered= orally administered
LXR-α and ABCA1 in aortic and liver of= in aortic and liver tissues of
HUVECs cell stain= strain
2- Provide a clean version of the manuscript in which the changes are highlighted in green or yellow.
Response:
Thanks a lot for the suggestion, which are very helpful for improving our paper. We are really sorry for our carelessness and poor writing abilities and appreciated so much that the reviewer pointed it out, and otherwise it might mislead the readers once it is published. We did check again, in the revised manuscript, we carefully carry out modifications.
Round 2
Reviewer 1 Report
Comments and Suggestions for Authors
Dear Authors,
It is very kind of you to make the adjustments and corrections. However, information about the analyses is still required. I quote your response to explain the rationale for making the adjustments:
"Multi-factor ANOVA method combined with Shapiro-Wilk and Wilks' Lambda and so on, and the corrections were shown in Fig. 2, Fig. 3, Fig. 4, Fig. 5, and Fig. 6 in the revised manuscript."
For the results associated with the figures as mentioned above, for example, Fig. 2: please indicate the Wilks' Lambda value from the MANOVA and the coefficient of determination (r2), which are the estimated values resulting from the multivariate effect for hypothesis testing. You should also add the effect size (partial Eta squared). This information can be included in the figure caption, which allows the reader to understand the analysis process and correctly reproduce this type of statistical modeling.
These points would be what remains to be addressed in your manuscript.
Best regards,
Author Response
Dear editors and reviewers:
Thank you for your letter and for the reviewers’ comments concerning our manuscript entitled " Citri Grandis exocarpium extract alleviate atherosclerosis in ApoE-/- mice by modulating the expression of TGF-β1, PI3K, AKT1, PPAR-γ, LXR-α and ABCA1" (ID: foods-3874678). Those comments are all valuable and very helpful for improving our paper, as well as the important guiding significance to our researches. We had carefully read the comments and made revisions which we hope to meet with approval. Revised portions were marked in yellow in the paper. The main corrections in the paper and the responds to the reviewer’s comments are as following.
Reviewer: 1
It is very kind of you to make the adjustments and corrections. However, information about the analyses is still required. I quote your response to explain the rationale for making the adjustments:
- "Multi-factor ANOVA method combined with Shapiro-Wilk and Wilks' Lambda and so on, and the corrections were shown in Fig. 2, Fig. 3, Fig. 4, Fig. 5, and Fig. 6 in the revised manuscript."
- For the results associated with the figures as mentioned above, for example, Fig. 2: please indicate the Wilks' Lambda value from the MANOVA and the coefficient of determination (r2), which are the estimated values resulting from the multivariate effect for hypothesis testing. You should also add the effect size (partial Eta squared). This information can be included in the figure caption, which allows the reader to understand the analysis process and correctly reproduce this type of statistical modeling.
These points would be what remains to be addressed in your manuscript.
Best regards,
Response:
You raised excellent comments. According to the suggested, we revisited the analysis method of the experimental raw data. Following the principles and criteria for dependent variable selection in MANOVA, however, there are some indexes, which include Fig.1-C (Cell viability), 1-D (Cell damage) and 1-E (Intracellular lipid droplets stained with ORO), Fig.2-C (serum ALT and AST), Fig.3-A (en face aortas oil red O stain), 3-B (Aortic root oil red O stain) and 3-C (Aortic root Masson stain), Fig. 4-C (Liver Oil Red O stain), 4-D (Liver Masson stain), should be analyzed using One-Way ANOVA method, the effect size (partial Eta squared) was added back again in the figure caption in the revised manuscript.
In addition, the other experimental data, such as Fig.1-F, 1-G, 1-H and 1-I (Cell migration and invasion), Fig.2-B (lipids four), Fig.5 (PPAR-γ, LXR-α, ABCA1 TGF-β, PI3K, AKT1 protein expression levels in aorta), Fig.6 (PPAR-γ, LXR-α, ABCA1 TGF-β, PI3K, AKT1 protein and mRNA expression levels in liver), were performed using the MANOVA method combined with a Wilk’s Lambda, etc., the Wilks' Lambda value from the MANOVA and the coefficient of determination (r2) also was added back in the figure caption in the revised manuscript.
Thank you for the kind reminds. In the revised manuscript, we carefully carry out modifications.

Reviewer 4 Report
Comments and Suggestions for Authors
Dear Authors,
As mentioned in the previous review.
Please provide a clean version when the changes only are highlighted in yellow.
Please provide a clean version and a tracked changes version.
Comments on the Quality of English Language
The plagiarism score is now around 27% which is high.
Review the clean version and avoid plagiarism.
Author Response
As mentioned in the previous review.
- Please provide a clean version when the changes only are highlighted in yellow.
- Please provide a clean version and a tracked changes version.
- The plagiarism score is now around 27% which is high. Review the clean version and avoid plagiarism.
Response:
Thanks a lot for the suggestion. These suggestions are very helpful for improving our paper. It’s important to avoid plagiarism. We again carried out reduction of the repetition rate and revision in the revised manuscript. In addition, we provide a clean version in which the revised portions were marked in bright-yellow in the paper.
Finally, fully acknowledge all positive points from the reviewers.
Sincerely,
Ping Xiong

Round 3
Reviewer 4 Report
Comments and Suggestions for Authors
The revised version of the manuscript shows a clear improvement in structure and focus. However, several areas would benefit from refinement to enhance mechanistic clarity and translational relevance. The following can be addressed before acceptance.
- Check the language again. For example: Citri grandis exocarpium—belonging to the Citrus genus in the family Rutaceae, which is the nearly mature or immature dried. Here you first use a hyphen and then a comma? Also no need for “which”= Citri grandis exocarpium, belonging to the Citrus genus in the family Rutaceae, is the nearly mature or immature dried
- Clarify HPLC detection method. The text inconsistently describes a PDA detector with “excitation/emission” wavelengths. Specify whether the detector is PDA or fluorescence (FLD) and list the exact wavelengths used. Expand compound profiling. Currently, only naringin and rhoifolin are quantified. Include chromatograms showing the full chemical fingerprint and at least 5–10 marker peaks (other flavonoids/coumarins) or justify limiting quantification to two markers. Provide precision (intra-/inter-day), recovery, and LOD/LOQ results in a supplementary table.
- CGE doses (0.4–1.6 g/kg) are high for a water extract. Add rationale and human-equivalent dose (HED) calculation. Cite prior toxicity or pharmacokinetic data.
- Clarify if animals were randomized, by whom, and whether outcome assessments were blinded to experimenter.
- Describe the vehicle composition and confirm it matched across all groups.
- Justify simvastatin dose and confirm vehicle equivalence.
- CGE used at 1.25–5 mg/mL in HUVECs seems supraphysiological. Provide solubility and cytotoxicity range data (IC₅₀) and justify concentration choice.
- Describe how fields were selected for the quantification of the migration assays, analysis method, software used, and include number of fields counted per well and total replicates.
- I think that MANOVA/Wilks’ test since many single-variable comparisons do not require multivariate tests. Check this. Clarify where MANOVA was used; apply one-way ANOVA with appropriate post hoc tests for single endpoints.
- Tone down overstated causality noting that pathway changes (mRNA/protein) are correlative. Rephrase “CGE regulates” to “CGE may modulate” or “is associated with changes in” TGF-β1/PI3K/AKT1 and PPAR-γ/LXR-α/ABCA1 pathways.
- If feasible, add an inhibitor experiment (e.g., PPAR-γ antagonist GW9662) or cholesterol efflux assay to confirm mechanistic claims.
- Add scale bars to all histological, ORO and all images.
Author Response
Response to Reviewers’ comments
Dear reviewers and editors:
Thank you for your letter and for the reviewers’ comments concerning our manuscript entitled " Citri Grandis exocarpium extract alleviate atherosclerosis in ApoE-/- mice by modulating the expression of TGF-β1, PI3K, AKT1, PPAR-γ, LXR-α and ABCA1" (ID: foods-3874678). Those comments are all valuable and very helpful for improving our paper, as well as the important guiding significance to our researches. We had carefully read the comments and made revisions which we hope to meet with approval. Revised portions were marked in yellow in the paper. The main corrections in the paper and the responds to the reviewer’s comments are as following.
Reviewer 4
The revised version of the manuscript shows a clear improvement in structure and focus. However, several areas would benefit from refinement to enhance mechanistic clarity and translational relevance. The following can be addressed before acceptance.
- Check the language again. For example: Citri grandis exocarpium—belonging to the Citrus genus in the family Rutaceae, which is the nearly mature or immature dried. Here you first use a hyphen and then a comma? Also no need for “which”= Citri grandis exocarpium, belonging to the Citrus genus in the family Rutaceae, is the nearly mature or immature dried
Response:
Thanks a lot for the suggestion. The reviewer read so carefully about our manuscript and thanks for your time! In the revised manuscript, the sentence “Citri grandis exocarpium (C.grandis)—belonging to the Citrus genus in the family Rutaceae, which is the nearly mature or immature dried exocarp of this species” was changed to “Citri grandis exocarpium (C.grandis)—belonging to the Citrus genus in the family Rutaceae, is the nearly mature or immature dried exocarp of this species. In addition, we check the language again, and revised portions were marked in yellow.
2.Clarify HPLC detection method. The text inconsistently describes a PDA detector with “excitation/emission” wavelengths. Specify whether the detector is PDA or fluorescence (FLD) and list the exact wavelengths used. Expand compound profiling. Currently, only naringin and rhoifolin are quantified. Include chromatograms showing the full chemical fingerprint and at least 5–10 marker peaks (other flavonoids/coumarins) or justify limiting quantification to two markers. Provide precision (intra-/inter-day), recovery, and LOD/LOQ results in a supplementary table.
Response:
We sincerely apologize for our mistakes and for the confusing caused to you. In the study, PAD detector should be corrected to “DAD detector”, that is, Waters alliance e2695 HPLC system with DAD detector. In addition, chromatographic fingerprints were recorded at a detection wavelength of 320 nm. The description on “PDA detector with an emission wavelength of 410 nm and an excitation wavelength of 320 nm” is incorrect. We appreciated so much that the reviewer pointed it out, otherwise it might mislead the readers once it published. In this study, we performed the fingerprint spectrum analysis for this batch of CGE dried extract powder, the HPLC specific chromatogram characterization of CGE was shown in Fig.1. A total of 10 chromatographic peaks were labeled and compared with reference standards. It was confirmed that peaks 4, 5, 6, and 9 were rhoifolin, naringin, meranzin hydrate, and lsomerazin, respectively.
Fig.1. HPLC fingerprint of CGE dried extract powder (The picture can be found in the PDF version of the manuscript and the author's reply.)
Where peak 4 is rhoifolin; peak 5 is naringin; peak 6 is meranzin hydrate; peak 9 is lsomerazin.
The standard curve was draw with the concentration of reference solution as the x-axis (X), and the peak area of the reference spectrum as the y-axis (Y). Through linear regression analysis, the linear regression equation for the content of rhoifolin is Y= 34256 X+ 638.65 (γ = 0.9999), and the linear regression equation for the content of naringin is Y = 5225.1 X +16157 (γ = 0.9999).
Based on the above regression equation and combined with the peak area of the chromatogram spectrum of this test sample, the content of naringin in this batch of C.grandis dried extract is 96.34 mg/g, and the content of rhoifolin is 8.86 mg/g.
We only provided limiting quantification to the two markers of naringin and rhoifolin, the reason
dues to that they are specified key quality control indicators in Chinese pharmacopoeia. According to the Chinese Pharmacopoeia 2025 edition, the content of naringin in dried product should not be less than 3.5%.
In addition, to control C.grandis product quality, we often need to quantify its water extract, so we established the HPLC quantitative detection method and its methodological validation. In this study, to determine the accurate dosage of experimental animal, we used dried extract powder of C.grandis, and the quantitative analysis utilized the established detection methods. But we did not conduct the methodological validation for the batch of dried powder of CGE. The aim of this study was to determine whether CGE alleviates atherosclerosis in apoE-/- mice and its possible mechanism.
In addition, as suggested by editorial teams:“the representative HPLC chromatograms of CGE can be presented as Supplementary data” in the second revised draft.
- CGE doses (0.4–1.6 g/kg) are high for a water extract. Add rationale and human-equivalent dose (HED) calculation. Cite prior toxicity or pharmacokinetic data.
Response:
Thanks a lot for the nice question. According to the body surface area, human-equivalent dose (HED) was calculated with the equivalent coefficient conversion algorithm (Table 1).
Table 1 Equivalent dose ratio of different experimental animals to humans (60kg) (mg·kg-1)
|
Dose ratio |
Different experimental animals |
||||||
|
Mouse |
Rat |
Guinea pig |
Rabbit |
Cat |
Monkey |
Dog |
|
|
9.1 |
6.3 |
5.42 |
3.27 |
2.73 |
1.05 |
1.87 |
|
The calculation process is as follows:
Human-equivalent dose (HED) of CGE = the mice dose of CGE ÷ 9.1
Therefore,
- High dose of CGE daily for human = (1.6g/kg BW) ÷ 9.1×60kg· BW = 0.1758 g/kg ×60kg· BW =10.55g/d
- Medium dose of CGE daily for human = (0.8 g/kg BW) ÷ 9.1 ×60kg· BW = 0.0879 g /kg ×60kg· BW = 5.27 g/d
- Low dose of CGE daily for human = (0.4 g/kg BW) ÷ 9.1×60kg· BW = 0.0439 g/kg ×60kg· BW = 2.64 g/d
Actually, the total yield of CGE was 29.24% in our extraction, that is, a total of 1g of CGE is equivalent to 3.42g of C. grandis. However, in our study, CGE is crude water extract, which was not separated and purified, so some inactive impurities, such as protein, starch, pigment and
monosaccharide and so forth, are possibly still present. In addition, as a homologous plant in medicine and food, it had been confirmed that the no observed adverse effect level (NOAEL) of
- grandis water extract powder for rat was 19.5 g/kg/d body weight (BW) [1-2], that is, the NOAEL dose of CGE daily for human 19.5 g/kg· BW÷6.3×60kg · BW = 185.71 g/d.
HED of CGE = the rat dose of CGE ÷ 6.3
References:
- LI Xin, WANG Ping, PENG Baoying, WANG Fengyan, PENG Jiewen,LONG Chaoyang, LIANG Yuanjin, HUANG Zhibiao. Food safety assessment of Citrus grandis fructus immaturus. South China J Prev Med, 2025, 51(4): 408-412.
- PENG Bao-ying, TANG Li, WANG Feng-yan, ZHANG Zi-hong, ZHOU Yi-lin,CHEN Mei-fen, HUANG Zhi-biao, LI Xin. Toxicological safety evaluation on Citrus grandis fructus immaturus. J Toxicol. 2024, 38 (6): 461-466.
- Clarify if animals were randomized, by whom, and whether outcome assessments were blinded to experimenter.
Response:
Thanks a lot. In this study, 60 of apoE-/- C57BL/6J male mice were randomly divided into five groups except for 12 C57BL/6J male mice in normal control group, 12 mice for each group. Random grouping complied with the random number table method.
In vitro test, Jun-hui Zhao conducted the study on the dose–inhibition effect of CGE on HUVEC cells viability, and the protective effects of CGE on cholesterol (10 μM)-induced cell injury. Jing Xu conducted the study on the dose–response relationship of CGE on LDH release levels in the absence or presence of 10 μM cholesterol, and Lipid accumulation of HUVECs cultured with CGE for 24 in the presence or absence of 10μΜ cholesterol, anti-migration effects of CGE on ox-LDL-induced HUVEC cells, etc.
In vivo test, animal experimental randomized grouping, modeling, feeding administration and sample collection, etc were performed by Jun-hui Zhao, Jing Xu and Wen-zhao Wen. All samples were measured and analyzed by Jun-hui Zhao, Jing Xu and Wen-zhao Wen, while all test data in statistical analysis and outcome assessments was performed by Jun-rong Guo, Zhuo-ya Zhang and Ping Xiong. So the outcome assessments were blinded to experimenter.
- Describe the vehicle composition and confirm it matched across all groups.
Response:
Thanks a lot a good question. In our study, the vehicle composition is Dulbecco’s Modified Eagle Medium (DMEM) in cell experiment (in vitro), CGE was dissolved in DMEM medium and filtered through a 0.22 μ m membrane. In the animal experiment, the vehicle composition is sterilized filtered water (in vivo). These two vehicles are non-toxic, and matched across all groups.
- Justify simvastatin dose and confirm vehicle equivalence.
Response:
Thanks for the good suggestion. Actually, the clinical routine dosage of simvastatin is 10 mg/d. According to the body surface area, mice-equivalent dose (MED) was calculated with the equivalent coefficient conversion algorithm. However, some people can take 5mg/d or 2.5 mg/d
orally due to the state of illness and the intolerance to side effects of simvastatin (muscle pain, abnormal liver function, gastrointestinal discomfort, etc).
Table 2 Equivalent dose ratio of different experimental animals to humans (60kg) (mg·kg-1)
|
Dose ratio |
Different experimental animals |
||||||
|
Mouse |
Rat |
Guinea pig |
Rabbit |
Cat |
Monkey |
Dog |
|
|
9.1 |
6.3 |
5.42 |
3.27 |
2.73 |
1.05 |
1.87 |
|
The calculation process is as follows:
HED of simvastatin = the mice dose of simvastatin ÷ 9.1
Thus, the daily dose of simvastatin for mice = human daily dose (10mg/d) × 9.1 ÷ 60 kg · BW = 1.5617 mg/kg/d
In addition, according to some reported literature, as positive control group, mice were treated with either 2 mg/kg simvastatin per day or vehicle [1]; LI Wei et al. reported that 1.3 mg/kg/d of simvastatin was administrated to C57BL/6J mice as positive group [2].
References:
- Eric F Steinmetz, Celine Buckley, Murray L Shames, Terri L Ennis, Sarah J Vanvickle-Chavez, Dongli
Mao, Lee A Goeddel, Cherady J Hawkins, Robert W Thompson. Treatment with simvastatin suppresses the development of experimental abdominal aortic aneurysms in normal and hypercholesterolemic mice. Ann Surg. 2005; 241(1):92-101. doi: .1097/01.sla.0000150258.36236.e0.
2.LI Wei,LI Luyao,QU Liping,LIU Honglin,LAI Mengting,WANG Ziqian,ZOU Wenjun. Study
on the Effect and Mechanism of Di’ao Xinxuekang Combined with Simvastatin on Atherosclero⁃sis Mice.
Traditional Chinese drug research & clinical pharmacology. 2024, 35(6): 798-804.
- CGE used at 1.25–5 mg/mL in HUVECs seems supraphysiological. Provide solubility and cytotoxicity range data (IC₅₀) and justify concentration choice.
Response:
That’s a very nice question. The reviewer read so carefully about our manuscript and thanks for your time! In fact, CGE has good water-solubility, and 40mg of CGE could completely dissolve in 1mL of water.
Table 3The cytostatic effect of CGE on HUVECs cells(x̅±SE,n=6)
|
CGE (mg/mL) |
Cytostatic rate(%) |
|
1.25 |
-5.80±6.45 |
|
2.5 |
-1.24±4.89 |
|
5 |
21.48±8.40 |
|
10 |
81.69±8.164 |
|
20 |
92.435±0.67 |
Fig.2.Dose–inhibition curve of CGE on HUVEC cells viability. (The picture can be found in the PDF version of the manuscript and the author's reply.)
As was shown in table 3, 1.25、2.5 mg/mL of CGE could promote the proliferation of HUVEC cells in vitro. However, HUVEC cell viability decreased at a dose-dependent manner with the further increase of CGE concentration. To further study the protective effects of CGE on cholesterol (10μM)-induced cell injury, so we select the three concentration with no obvious inhibitory effect of 1.25、2.5 and 5 mg/mL, and the cytotoxicity IC50 value = 6.94 mg/mL.
- Describe how fields were selected for the quantification of the migration assays, analysis method, software used, and include number of fields counted per well and total replicates.
Response:
Thanks a lot. The experimental operations follow the procedure below. In brief, the HUVEC cells (3×105/well) were inoculated in a 6-well cell plate and treated with 50 µg/mL ox-LDL and 1.25, 2.50, and 5.00 mg/mL of CGE (final concentration in the well) for 24 h at 37℃. In addition, the transwell was coated with 0.1% gelatin for 30 min at 37℃, following by washing with PBS three times. Then, cell suspension were incubated in the upper chamber of a 24-well transwell chamber (JET BIOFIL Co., Ltd., Guangzhou, China) at the density of 5×104/well with DMEM (10% FBS) in a final volume of 0.1mL per well, while the bottom chambers were filled with the DMEM with 10% FBS. After the incubation for 24h at 37℃, cells were removed away from the upper side of the membrane, and cells on bottom side were seen as the invasion cells. Then, the invasion cells were fixed with 4% PFA at RT for 30min. The invasion cells were washed with PBS three times and then stained with 0.1% crystal violet solution for 15 min. Finally, random five fields of view per well was acquired under a microscope (ECLIPSE Nikon, Tokyo, Japan), and the number of positive cells in each field of view was counted using the Image J software (version: 1.54j, NIH, MD, USA). The experiment was repeated three times. The SPSS statistical Software package (SPSS Inc. Chicago, IL, USA) with Windows Version 20.0 was applied for statistical analyses of the experiment data. The statistical results are presented as ± SE. The comparison of the measured variables between groups was performed using One-Way ANOVA method combined with a t-test and so on. If the variance is homogeneous, the LSD test for multiple comparisons should be followed by that analysis, if not, Tamhane’s T2 (M) and Dunnett’s T3 test should be combined. Levels of significance were set at P < 0.05, P < 0.05 indicated that the difference was statistically significant, and P < 0.01 indicated that the statistical difference was extremely significant.
- I think that MANOVA/Wilks’ test since many single-variable comparisons do not require multivariate tests. Check this. Clarify where MANOVA was used; apply one-way ANOVA with appropriate post hoc tests for single endpoints.
Response:
Thanks a lot. In fact, the comparison of measured variables between groups was performed using One-Way ANOVA method combined with a t-test and so on in our first manuscript. We also think that one-way ANOVA with appropriate post hoc tests is appropriate.
As described in our first manuscript: “The comparison of the measured variables between groups was performed using One-Way ANOVA method combined with a t-test and so on. If the variance is homogeneous, the LSD test for multiple comparisons should be followed by that analysis, if not, Tamhane’s T2 (M) and Dunnett’s T3 test should be combined. Levels of significance were set at P < 0.05, P < 0.05 indicated that the difference was statistically significant, and P < 0.01 indicated that the statistical difference was extremely significant”.
However, the comments and suggestions of reviewer 1 in Round 1 pointed out:
“Dear Authors,
The study is very interesting and contains results that will be useful for this field of knowledge. However, before publication, several points related to reproducibility and repeatability must be adjusted and corrected. The section I specifically reviewed is on statistical analyses, and this is evidently reflected in the results.
- The mean values should be graphed with standard error bars, because inferential parametric tests were applied (see Cumming, G., Fidler, F., & Vaux, D. L. (2007). Error bars in experimental biology. The Journal of Cell Biology, 177(1), 7-11).
- These tests (one-way ANOVA and independent samples t-test) require three assumptions: data independence, compliance with the normal error distribution, and homogeneity of variances; the latter indicates that they were verified. However, they must indicate what type of hypothesis test was applied. Likewise, with the normal error distribution, indicate the test and statistic values.
3. They applied more than 60 one-way ANOVAs, which inflates the probability of committing type II statistical errors. In this sense, it is advisable to perform MANOVAs with a matrix of 18 rows (n=18) by 18 columns (response variables at most), so as not to violate the assumption of symmetry of the data matrix. In this sense, the minimum number of response variables must be 5 for the MANOVA. Therefore, they must readjust their analyses. - Applying this type of analysis requires not violating the assumptions of normal error distribution and homogeneity of variances. Therefore, there are several options: transform the data to the base 10 logarithm, or rank (Conover, W. J., & Iman, R. L. (1981). Rank transformations as a bridge between parametric and nonparametric statistics. The American Statistician, 35(3), 124-129).
Reanalysis is essential to minimize type II statistical errors. It is common practice to apply too many independent tests, and this deficiency can be resolved with MANOVA.
Best regards,”
So according to the suggestions of reviewer 1, some experimental data, such as Fig.1-F, 1-G, 1-H and 1-I (Cell migration and invasion), Fig.2-B (lipids four), Fig.5 (PPAR-γ, LXR-α, ABCA1 TGF-β, PI3K, AKT1 protein expression levels in aorta), Fig.6 (PPAR-γ, LXR-α, ABCA1 TGF-β, PI3K, AKT1 protein and mRNA expression levels in liver), were performed using the MANOVA method combined with a Wilk’s Lambda, etc., the Wilks' Lambda value from the MANOVA and the coefficient of determination (r2) also was added back in the figure caption in the revised manuscript.
For the above described reasons, the dependent variables that cannot be grouped together for MANOVA analysis were analyzed using one-way ANOVA. Moreover, in the data analysis process, this study conducted one-way ANOVA and post hoc testing on all variables. In response to the previous reviewer's suggestion, we also conducted a multivariate analysis of variance for cross validation. Both methods indicate significant differences between groups, and the conclusion keeps consistent, which enhances the stability of the research results.
Anyway, the suggestions of both experts are very good and very helpful for improving our paper.
- Tone down overstated causality noting that pathway changes (mRNA/protein) are correlative. Rephrase “CGE regulates” to “CGE may modulate” or “is associated with changes in” TGF-β1/PI3K/AKT1 and PPAR-γ/LXR-α/ABCA1 pathways.
Response:
Thanks a lot for the suggestion. We fully agree with the expert’s viewpoint. In the conclusion section, lines 5-6, “These effects of CGE are likely correlated to the TGF-β/PI3K/ALT1 and PPARγ/LXRα/ABCA1 signaling pathways.” The causality has been toned down.
- If feasible, add an inhibitor experiment (e.g., PPAR-γ antagonist GW9662) or cholesterol efflux assay to confirm mechanistic claims.
Response:
That’s a very nice question. In our manuscript, we haven’t found the real targets of action, and identified only possible related pathways/proteins. In fact, these effects of CGE may target at a certain upstream factors determining PPAR-γ upregulation, such as AMPK (Adenosine Monophosphate-Activated Protein Kinase, AMP-activated protein kinase) or MAPK protein family (including TAK1), and so on. CGE possibly exert effects through not only one pathway.
Anyway, it is limited in current exploring the exact target research in our manuscript. It’s really true that there must be some hidden mechanisms associated with the anti-hyperlipidemia and anti-atherosclerosis. At present, there’s some way to go from understand definite mechanism due to the complex chemical compositions of CGE. We will continue working on the detailed mechanistic study of CGE, especially the effect target of key active ingredients (rhoifolin, naringin), in the near future since the complicacy might not be fully understood in one study. The focus of this manuscript is the identification of CGE in the prevention of hyperlipidemia and atherosclerosis, which open up the window for further investigations.
References:
- Li-juan Zhang, Shun-ying Li, Shao-li Fan, Aobulikasimu nuerbiye, Jiu-yang Zhao, Bahetiyaer Keremu, Lu
Yang, Ke Zhang, Hang-yu Wang, Jin-hui Wang. Hepatoprotective activity of mulberry extract in NAFLD mice for regulating lipid metabolism and inflammation identified via AMPK/PPAR-γ/ NF-κB axis. Journal of Ethnopharmacology. 2025, 351: 120108. doi: 10.1016/j.jep.2025.120108.
- Young-Seo Yoon, Kyung-Sook Chung, Su-Yeon Lee, So-Won Heo, Ye-Rin Kim, Jong Kil Lee, Hyunjae Kim, Soyoon Park, Yu-Kyong Shine and Kyung-Tae Lee. Anti-obesity effects of a standardized ethanol extract of Eisenia bicyclis by regulating the AMPK signaling pathway in 3T3-L1 cells and HFD-induced mice. Food and function. 2024, 15(12):6424-6437.doi: 10.1039/d4fo00759j.
- XIA Xuemei, CHEN Yan. Mechanism of TAK1 regulating insulin resistance and lipid metabolism in hepatocytes through PPARγ. Zhejiang journal of integrated tradition Chinese and western medicine. 2025, 35 (9): 791-796.
- WU Xize, YU Kaifeng, PAN Xue, PAN Jiaxiang, HUANG Yuxi, WANG Ruiying. Mechanism of Didang Decoction in promoting reverse cholesterol transport through the AMPK/PPARγ/LXRα signaling axis in the treatment of atherosclerosis. China journal of traditional Chinese medicine and pharmacy. 2025, 40(5): 2481-2489.
- Add scale bars to all histological, ORO and all images.
Response:
Thanks for the good suggestion. As the expert suggested, we had added scale bars to all histological, oil red O staining and all images in the revised manuscript.
Finally, fully acknowledge all positive points from the reviewer.
